# Making the most of your day: online learning for optimal allocation of time

**Etienne Boursier**
Centre Borelli, ENS Paris-Saclay, France
`etienne.boursier1@gmail.com`

**Tristan Garrec**
Centre Borelli, ENS Paris-Saclay, France
EDF Lab, Palaiseau, France
`tristan.garrec@ut-capitole.fr`

**Vianney Perchet**
CREST, ENSAE Paris, France
Criteo AI Lab, Paris, France
`vianney@ensae.fr`

**Marco Scarsini**
Department of Economics and Finance, LUISS, Rome
`marco.scarsini@luiss.it`

## Abstract

We study online learning for optimal allocation when the resource to be allocated is time. An agent receives task proposals sequentially according to a Poisson process and can either accept or reject a proposed task. If she accepts the proposal, she is busy for the duration of the task and obtains a reward that depends on the task duration. If she rejects it, she remains on hold until a new task proposal arrives. We study the regret incurred by the agent, first when she knows her reward function but does not know the distribution of the task duration, and then when she does not know her reward function, either. This natural setting bears similarities with contextual (one-armed) bandits, but with the crucial difference that the normalized reward associated to a context depends on the whole distribution of contexts.

## 1 Introduction

**Motivation.** A driver filling her shift with rides, a landlord renting an estate short-term, an independent deliveryman, a single server that can make computations online, a communication system receiving a large number of calls, etc. all face the same trade-off. There is a unique resource that can be allocated to some tasks/clients for some duration. The main constraint is that, once it is allocated, the resource becomes unavailable for the whole duration. As a consequence, if a "better" request arrived during this time, it could not be accepted and would be lost. Allocating the resource for some duration has some cost but generates some rewards – possibly both unknown and random. For instance, an estate must be cleaned up after each rental, thus generating some fixed costs; on the other hand, guests might break something, which explains why these costs are both random and unknown beforehand. Similarly, the relevance of a call is unknown beforehand. Concerning duration, the shorter the request the better (if the net reward is the same). Indeed, the resource could be allocated twice in the same amount of time.

The ideal request would therefore be of short duration and large reward; this maximizes the revenue per time. A possible policy could be to wait for this kind of request, declining the other ones (too long and/or less profitable). On the other hand, such a request could be very rare. So it might be more rewarding in the long run to accept any request, at the risk of "missing" the ideal one.

35th Conference on Neural Information Processing Systems (NeurIPS 2021).

Some clear trade-offs arise. The first one is between a greedy policy that accepts only the highest profitable requests – at the risk of staying idle quite often – and a safe policy that accepts every request – but unfortunately also the non-profitable ones. The second trade-off concerns the learning phase; indeed, at first and because of the randomness, the actual net reward of a request is unknown and must be learned on the fly. The safe policy will gather a lot of information (possibly at a high cost) while the greedy one might lose some possible valuable information for the long run (in trying to optimize the short term revenue).

We adopt the perspective of an agent seeking to optimize her earned income for some large duration. The agent receives task proposals sequentially, following a Poisson process. When a task is proposed, the agent observes its expected duration and can then either accept or reject it. If she accepts it, she cannot receive any new proposals for the whole duration of the task. At the end of the task she observes her reward, which is a function of the duration of the task. If, on the contrary, she rejects the task, she remains on hold until she receives a new task proposal.

The agent's policies are evaluated in terms of their expected regret, which is the difference between the cumulative rewards obtained until $T$ under the optimal policy and under the implemented agent policy (as usual the total length could also be random or unknown (Degenne and Perchet, 2016)). In this setting, the "optimal" policy is within the class of policies that accept – or not – tasks whose length belongs to some given acceptance set (say, larger than some threshold, or in some specific Borel subset, depending on the regularity of the reward function).

**Organization and main contributions.** In Section 2, we formally introduce the model and the problem faced by an oracle who knows the distribution of task durations as well as the reward function (quite importantly, we emphasize again that the oracle policy must be independent of the realized rewards). Using continuous-time dynamic programming principles, we construct a quasi-optimal policy in terms of accepted and rejected tasks: this translates into a single optimal threshold for the ratio of the reward to the duration, called the profitability function. Any task with a profitability above this threshold is accepted, and the other ones are declined. As a benchmark, we first assume in Section 3 that the agent knows the reward function $r(\cdot)$, but ignores the distribution of task durations. The introduced techniques can be generalized, in the following sections, to further incorporate estimations of $r(\cdot)$. In that case, our base algorithm has a regret scaling as $\mathcal{O}(\sqrt{T})$; obviously, this cannot be improved without additional assumptions, ensuring minimax optimality. In Section 4, the reward function is not known to the agent anymore and the reward realizations are assumed to be noisy. To get non-trivial estimation rates, regularity – i.e., $(L, \beta)$-Hölder – of the reward function is assumed. Modifying the basic algorithm to incorporate non-parametric estimation of $r$ yields a regret scaling as $\mathcal{O}(T^{1-\eta}\sqrt{\ln T})$ where $\eta = \beta/(2\beta + 1)$. As this is the standard error rate in classification (Tsybakov, 2006), minimax optimality (up to $\log$-term) is achieved again. Finally, our different algorithms are empirically evaluated on simple toy examples in Section 5. Due to space constraints, all the proofs are deferred to the Appendix.

**Related work.** As data are gathered sequentially, our problem bears similarities with online learning and multi-armed bandit (Bubeck and Cesa-Bianchi, 2012). The main difference with multi-armed bandit or resource allocation problems (Fontaine et al., 2020) is that the agent's only resource is her time, which has to be spent wisely and, most importantly, saving it actually has some unknown value. The problem of time allocation actually goes way back. In economic theory, Becker's seminal paper (Becker, 1965) evaluates the full costs of non-working activities as the sum of their market prices and the forgone value of the time used to enjoy the activities. Various empirical works have followed (see, e.g., Juster and Stafford, 1991; Chabris et al., 2009).

The problem of online scheduling has been studied in several possible variations (see, e.g., Pruhs et al., 2004, for a survey). Various subsequent papers consider scheduling models that combines online and stochastic aspects (see, e.g., Megow et al., 2006; Chou et al., 2006; Vredeveld, 2012; Marbán et al., 2012; Skutella et al., 2016). Yet these models are different as they aim at minimizing the makespan of all received jobs, while actions here consist in accepting/declining tasks. In online admission control, a controller receives requests and decides on the fly to accept/reject them. However, the problem of admission control is more intricate since a request is served through some path, chosen by the controller, in a graph. Even solving it offline thus remains a great challenge (Wu and Bertsekas, 2001; Leguay et al., 2016).

An online model of reward maximization over time budget has been proposed by Cayci et al. (2019, 2020), where the agent does not observe the duration of a task before accepting it. As a consequence, the optimal policy always chooses the same action, which is maximizing the profitability. Since the duration of a task is observed before taking the decision in our problem, the decision of the optimal policy depends on this duration, used as a covariate. Cayci et al. (2019) thus aim at determining the arm with the largest profitability, while our main goal is here to estimate the profitability threshold from which we start accepting the tasks. The considered problems and proposed solutions thus largely differ in these two settings.

Our problem is strongly related to contextual multi-armed bandits, where each arm produces a noisy reward that depends on an observable context. Indeed, the agent faces a one arm contextual bandit problem (Sarkar, 1991), with the crucial difference that the normalized reward associated to a context depends on the whole distribution of contexts (and not just the current context). The literature on contextual bandits, also known as bandits with covariates, actually goes back to Woodroofe (1979) and has seen extensive contributions in different settings (see, e.g., Yang and Zhu, 2002; Wang et al., 2005; Rigollet and Zeevi, 2010; Goldenshluger and Zeevi, 2009; Perchet and Rigollet, 2013).

This problem is also related to bandits with knapsacks (see Slivkins, 2019, Chapter 10) introduced by Badanidiyuru et al. (2013) and even more specifically to contextual bandits with knapsacks (Badanidiyuru et al., 2014; Agrawal et al., 2016), where pulling an arm consumes a limited resource. Time is the resource of the agent here, while both time and resource are well separated quantities in bandits with knapsacks. Especially, bandits with knapsacks assume the existence of a null arm, which does not consume any resource. This ensures the feasibility of the linear program giving the optimal fixed strategy. Here, the null arm (declining the task) still consumes the waiting time before receiving the next task proposal. Bandits with knapsacks strategies are thus not adapted to our problem, which remains solvable thanks to a particular problem structure. The problem of online knapsacks is also worth mentioning (Noga and Sarbua, 2005; Chakrabarty et al., 2008; Han and Makino, 2009; Böckenhauer et al., 2014), where the resources consumed by each action are observed beforehand. This is less related to our work, as online knapsacks consider a competitive analysis, whereas we here aim at minimizing the regret with stochastic contexts/rewards.

## 2 Model and benchmark

### 2.1 The problem: allocation of time

All the different notations used in the paper are summarized at the beginning of the Appendix to help the reader. We consider an agent who sequentially receives task proposals and decides whether to accept or decline them on the fly. The durations of the proposed tasks are assumed to be i.i.d. with an unknown law, and $X_i$ denotes the duration of the $i$-th task. If this task is accepted, the agent earns some reward $Y_i$ with expectation $r(X_i)$. The profitability is then defined as the function $x \mapsto r(x)/x$. We emphasize here that $Y_i$ is not observed before accepting (or actually completing) the $i$-th task and that the expected reward function $r(\cdot)$ is unknown to the agent at first. If task $i$ is accepted, the agent cannot accept any new proposals for the whole duration $X_i$. We assume in the following that the function $r$ is bounded in $[E, D]$ with $E \leq 0 \leq D$, and that the durations $X_i$ are upper bounded by $C$.

After completing the $i$-th task, or after declining it, the agent is on hold, waiting for a new proposal. We assume that idling times – denoted by $S_i$ – are also i.i.d., following some exponential law of parameter $\lambda$. This parameter is supposed to be known beforehand as it has a small impact on the learning cost (it can be quickly estimated, independently of the agent's policy). An equivalent formulation of the task arrival process is that proposals follow a Poisson process (with intensity $\lambda$) and the agent does not observe task proposals while occupied. This is a mere consequence of the memoryless property of Poisson processes.

The agent's objective is to maximize the expected sum of rewards obtained by choosing an appropriate acceptance policy. Given the decisions $(a_i)_{i \geq 1}$ in $\{0, 1\}$ (decline/accept), the total reward accumulated by the agent after the first $n$ proposals is equal to $\sum_{i=1}^{n} Y_i a_i$ and the required amount of time for this is $\mathcal{T}_n := \sum_{i=1}^{n} S_i + X_i a_i$. This amount of time is random and strongly depends on the policy. As a consequence, we consider that the agent optimizes the cumulative reward up to time $T$, so that the number of received tasks proposals is random and equal to $\theta := \min\{n \in \mathbb{N} \mid \mathcal{T}_n > T\}$. Mathematically, a policy of the agent is a function $\pi$ – taking as input a proposed task $X$, the

time $t$ and the history of observations $\mathcal{H}_t := (X_i, Y_i a_i)_{i:\mathcal{T}_i < t}$ – returning the decision $a \in \{0, 1\}$ (decline/accept). In the following, the optimal policy refers to the strategy $\pi$ maximizing the expected reward $U_\pi$ at time $T$, given by

$$U_\pi(T) = \mathbb{E}\left[\sum_{n=1}^{\theta} r(X_n)\pi(X_n, t_n, \mathcal{H}_{t_n})\right], \tag{2.1}$$

where $t_n := S_n + \sum_{i=1}^{n-1} S_i + X_i a_i$ is the time at which the $n$-th task is proposed.

We allow a slight boundary effect, i.e., the ending time can slightly exceed $T$, because the last task may not end exactly at time $T$. A first alternative would be to compute the cumulative revenue over completed tasks only, and another alternative would be to attribute the relative amount of time before $T$ spent on the last task. Either way, the boundary effect is of the order of a constant and has a negligible impact as $T$ increases.

## 2.2 The benchmark: description of the optimal policy

The benchmark to which the real agent is compared is an oracle who knows beforehand both the distribution of tasks and the reward function. It is easier to describe the reward of the optimal policy than the policy itself. Indeed, at the very last time $T$, the agent cannot accumulate any more reward, hence the remaining value is 0. We then compute the reward of the policy backward, using continuous dynamic programming principles. Formally, let $v(t)$ denote the "value" function, i.e., the expected reward that the optimal policy will accumulate on the remaining time interval $[t, T]$ if the agent is on hold at this very specific time $t \in [0, T]$. As mentioned before, we assume the boundary condition $v(T) = 0$. The next proposition gives the dynamic programming equation satisfied by $v(t)$ (with the notation $(z)_+ = \max\{z, 0\}$ for any $z \in \mathbb{R}$), as well as a simple function approximating this value.

**Proposition 2.1.** *The value function $v$ satisfies the dynamic programming equation*

$$\begin{cases} v'(t) = -\lambda \mathbb{E}\left[(r(X) + v(t+X) - v(t))_+\right] & \text{for all } t < T, \\ v(t) = 0 & \text{for all } t \geq T. \end{cases} \tag{2.2}$$

*If $X \leq C$, then its solution is bounded by the affine function $w : t \mapsto c^*(T-t)$ for any $t \in [0, T]$ as follows*

$$w(t - C) \geq v(t) \geq w(t), \tag{2.3}$$

*where $c^*$ is the unique root of the function*

$$\Phi : \begin{array}{l} \mathbb{R}_+ \to \mathbb{R} \\ c \mapsto \lambda \mathbb{E}\left[(r(X) - cX)_+\right] - c. \end{array} \tag{2.4}$$

The constant $c^*$ represents the optimal reward per time unit, hereafter referred to as the *optimal profitability threshold*. The function $w$ is the value function of the optimal policy when neglecting boundary effects. The proof of Proposition 2.1 largely relies on the memoryless property of the idling times and determining a benchmark policy without this memoryless assumption becomes much more intricate. Based on this, it is now possible to approach the optimal policy by only accepting task proposals with profitability at least $c^*$. This results in a stationary policy, i.e., its decisions depend neither on the time nor on past observations but only on the duration of the received task.

**Theorem 2.2.** *The policy $\pi^*$ which accepts a task with duration $x$ if and only if $r(x) \geq c^*x$ is $c^*C$-suboptimal, i.e., $U_{\pi^*} \geq \max_\pi U_\pi - c^*C$.*

In the considered model, it is thus possible to compute a quasi-optimal online policy.

## 2.3 Online learning policies and regret

Below we investigate several contexts where the agent lacks information about the environment, such as the task durations distribution and/or the (expected) reward function. As mentioned before, the objective of the agent is to maximize the cumulative reward gathered until $T$ (up to the same boundary effect as the optimal policy) or equivalently to minimize the policy regret, defined for the policy $\pi$ as

$$R(T) = c^*T - U_\pi(T). \tag{2.5}$$

Note that $R(T)$ is the difference between the expected rewards of strategies $\pi^*$ and $\pi$, up to some constant term due to the boundary effect.

# 3 Warm up: known reward, unknown distribution

First, we assume that the reward function $r$ is known to the agent, or equivalently, is observed along with incoming task proposals. However, the agent does not know the distribution $F$ of task durations and we emphasize here that the agent does not observe incoming task proposals while occupied with a task (though the converse assumption should only affect the results by a factor $C^{-1}$, the inverse of the maximal length of a task). We now define

$$\Phi_n : c \mapsto \lambda \frac{1}{n} \sum_{i=1}^{n} (r(X_i) - cX_i)_+ - c, \tag{3.1}$$

which is the empirical counterpart of $\Phi$. Moreover, let $c_n$ be the unique root of $\Phi_n$. One has $\mathbb{E}[\Phi_n(c)] = \Phi(c)$ for all $c \geq 0$ and all $n \geq 1$.

**Proposition 3.1.** *For all $\delta \in (0, 1]$ and $n \geq 1$, $\mathbb{P}\left(c_n - c^* > \lambda(D - E)\sqrt{\frac{\ln(1/\delta)}{2n}}\right) \leq \delta$.*

*Remark* 3.2. Proposition 3.1 states that the error in the estimation of $c^*$ scales with $\lambda(D - E)$. Notice that if the reward function is multiplied by some factor (and $\lambda$ is fixed), then $c^*$ is multiplied by the same factor; as a consequence, a linear dependency in $D - E$ is expected.
Similarly, since $c^* = \lambda \mathbb{E}\left[(r(X) - c^*X)_+\right]$, a small variation of $\lambda$ induces a variation of $c^*$ of the same order. As a consequence, a linear dependency in $\lambda$ (for small values of $\lambda$) is expected. And as both effects are multiplied, the dependency in $\lambda(D - E)$ is correct.
On the other hand, as $\lambda$ goes to infinity, $c^*$ converges to $\max_x r(x)/x$, so our deviation result seems irrelevant (for a small number of samples). This is due to the fact that for large $\lambda$, $c^*$ is not really the expectation of a random variable, but its essential supremum. In the proof, this appears when $\Phi(c^* + \varepsilon)$ is bounded by $-\varepsilon$. It is not difficult to see that a tighter upper-bound is

$$-\varepsilon(1 + \lambda q_\varepsilon), \quad \text{with} \quad q_\varepsilon := \mathbb{E}[X \mathbb{1}(r(X) \geq (c^* + \varepsilon)X)]. \tag{3.2}$$

Hence the approximation error scales with $\lambda(D - E)/(1 + \lambda q_\varepsilon)$ as $\lambda$ goes to infinity. However, this does not give explicit confidence intervals and the regime $\lambda \to \infty$ is not really interesting. As a consequence, the formulation of Proposition 3.1 is sufficient for our purposes.

The used policy is straightforward: upon receiving the $n$-th proposal, the agent computes $c_n$ and accepts the task $X_n$ if and only if $r(X_n) \geq c_n X_n$. The pseudo-code of this policy is given in Algorithm 1 below. The regret of this policy can be rewritten as

$$R(T) = c^* T - \mathbb{E}\left[\sum_{n=1}^{\theta} \gamma_n \mathbb{E}\left[X_n \mathbb{1}(r(X_n) \geq c_n X_n) + S_n\right]\right], \tag{3.3}$$

$$\text{where} \quad \gamma_n := \frac{\lambda \mathbb{E}[r(X_n) \mathbb{1}(r(X_n) \geq c_n X_n)]}{1 + \lambda \mathbb{E}[X_n \mathbb{1}(r(X_n) \geq c_n X_n)]} = \frac{\mathbb{E}[r(X_n) \mathbb{1}(r(X_n) \geq c_n X_n)]}{\mathbb{E}[S_n + X_n \mathbb{1}(r(X_n) \geq c_n X_n)]} \tag{3.4}$$

corresponds to the expected per-time reward obtained for the $n$-th task under the policy induced by $(c_n)_{n\geq 1}$. In the last expression, the numerator is indeed the expected reward per task while the denominator is the expected time spent per task (including the waiting time before receiving the proposal). Proposition D.1, postponed to the Appendix, gives a control of $\gamma_n$ entailing the following.

**Theorem 3.3.** *In the known reward, unknown distribution setting, the regret of Algorithm 1 satisfies*

$$R(T) \leq \lambda(D - E)C\sqrt{\frac{\pi}{2}}\sqrt{\lambda T + 1}. \tag{3.5}$$

*Remark* 3.4. Once again, one might question the dependency on the different parameters. As mentioned before, $\lambda(D - E)$ represents the scale at which $c^*$ grows and $C$ is the scale of $X$, so that $\lambda(D - E)C$ is the total global scaling of rewards. On the other hand, the parameter $\lambda$ also appears in the square-root term. This is because $\lambda T$ is approximately the order of magnitudes of the observed tasks.

*Remark* 3.5. The estimated profitability threshold $c_n$ can be efficiently approximated using binary search, since the function $\Phi_n$ is decreasing. The approximation error is ignored in the regret, as approximating $c_n$ up to, e.g., $n^{-2}$ only leads to an additional constant term in the regret.
The computation and memory complexities of Algorithm 1 are then respectively of order $n \log(n)$

---
**ALGORITHM 1:** Known reward algorithm
---
**input :** $r, \lambda$
$n = 0$
**while** $\sum_{i=1}^{n} S_i + X_i \mathbb{1}(r(X_i) \geq c_i X_i) < T$ **do**
    $n = n + 1$
    Wait $S_n$ and observe $X_n$
    Compute $c_n$ as the unique root of $\Phi_n$ defined in Equation (3.1)
    **if** $r(X_n) \geq c_n X_n$ **then** accept task and receive reward $r(X_n)$
    **else** reject task
**end**
---

and $n$ when receiving the $n$-th proposal, because the complete history of tasks is used to compute $c_n$, the root of $\Phi_n$. This can be improved by noting that only the tasks with profitability in

$$\left[ c_n - \lambda(D - E)\sqrt{\frac{\ln(1/\delta)}{2n}}, \ c_n + \lambda(D - E)\sqrt{\frac{\ln(1/\delta)}{2n}} \right]$$

need to be exactly stored, thanks to Proposition 3.1. Indeed, with high probability, tasks with smaller profitability do not contribute to $\Phi(c^*)$, while tasks with larger profitability fully contribute to $\Phi(c^*)$, meaning that only their sum has to be stored. As this interval is of length $1/\sqrt{n}$, the complexities of the algorithm become sublinear in $n$ under regularity assumptions on $X$ and the profitability function. The computational complexity can even be further improved to $\log(n)$ using binary search trees, as explained in Appendix B.

## 4 Bandit Feedback

We now assume that rewards are noisy and not known upfront. When the agent accepts a task $X_i$, the reward earned and observed over the period of time $X_i$ is $Y_i = r(X_i) + \varepsilon_i$, where $(\varepsilon_i)_{i \geq 1}$ is a sequence of $\sigma^2$-subgaussian independent random variables with $\mathbb{E}[\varepsilon_i] = 0$. If the task $X_i$ is rejected, then the associated reward is not observed; on the other hand, the agent is not occupied and thus may be proposed another task, possibly within the time window of length $X_i$.

To get non-trivial estimation rates, in the whole section we assume the reward function $r$ to be $(\beta, L)$-Hölder, whose definition is recalled below.

**Definition 4.1.** Let $\beta \in (0, 1]$ and $L > 0$. The function $r$ is $(\beta, L)$-Hölder if

$$|r(x) - r(x')| \leq L|x - x'|^\beta, \qquad \forall x, x' \in [0, C] \ .$$

The reward function $r(\cdot)$ is estimated by $\hat{r}_n(\cdot)$, constructed as a variant of a regressogram. The state space $[0, C]$ is partitioned regularly into $M$ bins $B_1, \ldots, B_M$ of size $h = C/M$. For every bin $B$, let $x^B := \min\{x \in B\}$. Similarly to Algorithm 1, $c^*$ is estimated by $\hat{c}_n$. To define the learning algorithm, we also construct an upper-estimate of $r(\cdot)$, denoted by $\hat{r}_n^+(\cdot)$, and a lower-estimate of $c^*$, denoted by $\hat{c}_n^-$.

The learning algorithm is actually quite simple. A task $X_n$ is accepted if and only if its best-case reward $\hat{r}_{n-1}^+(X_n)$ is bigger than the worst-case per-time value of its bin $\hat{c}_{n-1}^- x^{B(X_n)}$, where $X_n$ is in the bin $B(X_n)$. Notice that, if $\hat{r}_n^+(\cdot)$ is bigger than $r(\cdot)$ and $\hat{c}_n^-$ is always smaller than $c^*$, then any task accepted by the optimal algorithm (i.e., such that $r(X) \geq c^* X$) is accepted by the learning algorithm. Hence regret is incurred solely by accepting sub-optimal tasks. The question is therefore how fast tasks can be detected as sub-optimal, and whether declining them affects, or not, the estimation of $c^*$. The pseudo-code is given in Algorithm 2 at the end of this section.

We now describe the construction of $\hat{r}_n$ and $\hat{c}_n$ (as well as their upper and lower-estimates). Let $(X_1, Y_1, \ldots, X_n, Y_n)$ be the vector of observations of pairs (task, reward). We define for all bins $B$,

$$\hat{r}_n(x) = \hat{r}_n^B = \frac{1}{N_B} \sum_{i=1}^{n} Y_i \mathbb{1}(X_i \in B \text{ and } a_i = 1), \quad \forall x \in B$$

$$\text{with } N_B = \sum_{i=1}^{n} \mathbb{1}(X_i \in B \text{ and } a_i = 1).$$

(4.1)

We emphasize here that declined tasks are not used in the estimation of $r(\cdot)$ (as we shall see later, they are not used in the estimation of $c^*$, either). The upper estimate of $r(\cdot)$ is now defined as

$$\hat{r}_n^+(x) = \hat{r}_n(x) + \underbrace{\sqrt{\sigma^2 + \frac{L^2}{4}\left(\frac{C}{M}\right)^{2\beta}}\sqrt{\frac{\ln(M/\delta)}{2N_B}} + L\left(\frac{C}{M}\right)^{\beta}}_{:=\eta_{n-1}(x)}, \quad \forall x \in B. \qquad (4.2)$$

The following Lemma 4.2 states that $\hat{r}_n^+(\cdot)$ is indeed an upper-estimate of $r(\cdot)$.

**Lemma 4.2.** *For every $n \in \mathbb{N}$, we have $\mathbb{P}\left(\forall x \in [0, C], \ \hat{r}_n^+(x) \geq r(x)\right) \geq 1 - \delta$.*

It remains to construct $\hat{c}_n$. We define iteratively for every bin $B$

$$\tilde{r}_n^B = \begin{cases} 0 & \text{if a task in } B \text{ has ever been rejected,} \\ \hat{r}_n^B & \text{otherwise.} \end{cases} \qquad (4.3)$$

So $\tilde{r}_n$ is equal to the reward estimate $\hat{r}_n$, except on the eliminated bins that are known to be suboptimal, for which it is instead $0$. We then introduce the empirical counterpart of $\Phi$,

$$\hat{\Phi}_n : c \mapsto \lambda \sum_{j=1}^{M} \frac{N_{B_j}}{n} \left(\tilde{r}_n^{B_j} - cx^{B_j}\right)_+ - c. \qquad (4.4)$$

Let $\hat{c}_n$ be the unique root of $\hat{\Phi}_n$; the lower estimate of $c^*$ is then

$$\hat{c}_n^- = \hat{c}_n - \underbrace{\left(2\lambda\sqrt{\sigma^2 + \frac{(D-E)^2}{4}}\sqrt{\frac{\ln(1/\delta)}{n}} + \kappa\lambda\max\left(\sigma, \frac{D-E}{2}\right)\sqrt{\frac{\log(n)+1}{hn}} + \sqrt{8}\lambda\frac{Lh^\beta}{2^\beta} + \lambda^2 Dh\right)}_{:=\xi_{n-1}},$$

$$(4.5)$$

where $\kappa \leq 150$, is the square root of the universal constant introduced in (Györfi et al., 2002, Theorem 11.3). Although the following lemma looks trivial, it is actually a keystone of our main result. It states that the optimal profitability threshold may be computed solely based on the accepted tasks – no matter what the rewards associated to declined tasks are. This is of crucial relevance, since it implies that a learning algorithm may actually decline tasks that are clearly non-profitable, without degrading the quality and the speed of profitability threshold estimation.

**Lemma 4.3.** *Let $c^*$ be the optimal profitability threshold associated to $r$, i.e., $c^*$ is the unique root of $c \mapsto \Phi(c) = \lambda\,\mathbb{E}\left[(r(X) - cX)_+\right] - c$ and let $\mathcal{E} \subset \{x \in [0, C] \mid r(x) \leq c^*x\}$ be any subset of sub-profitable tasks.*
*Then the profitability threshold $\tilde{c}$ associated to the modified reward function $r_{\mathcal{E}}(x) := r(x)\,\mathbb{1}(x \notin \mathcal{E})$ is equal to $c^*$, or, stated otherwise,*

$$c^* = \lambda\,\mathbb{E}\left[(r_{\mathcal{E}}(X) - c^*X)_+\right], \quad \forall \mathcal{E} \subset \{x \in [0, C] \mid r(x) \leq c^*x\}.$$

This implies that $\hat{c}_n^-$ is indeed an under-estimate.

**Proposition 4.4.** *For all $N \in \mathbb{N}$, we have $\mathbb{P}(\forall n \leq N, \hat{c}_n^- \leq c^*) \geq 1 - 2N\delta$.*

We can now state our main result.

**Theorem 4.5.** *If $r$ is $(L, \beta)$-Hölder, then the regret of Algorithm 2, taking $\delta = 1/T^2$ and $M = \left\lceil CL^{\frac{2}{2\beta+1}}(\lambda T + 1)^{\frac{1}{2\beta+1}}\right\rceil$, satisfies*

$$R(T) \leq \kappa_1 \lambda C \max(\sigma, D - E)L^{\frac{1}{2\beta+1}}(\lambda T + 1)^{\frac{\beta+1}{2\beta+1}}\sqrt{\ln(\lambda T + 1)},$$

*where $\kappa_1$ is some universal constant independent of any problem parameter.*

A similar lower bound is proved in Appendix A.4, which implies that Algorithm 2 optimally scales with $T$, up to $\log$ terms. Even faster rates of convergence hold under additional assumptions on the task distribution or the profitability function and are provided in Appendix A for the cases of finite support of $X$, margin condition or monotone profitability function.

*Remark* 4.6. The complexity of Algorithm 2 when receiving a single task scales with the number of bins $M$, because the function $\hat{\Phi}_n$ uses the discretization over the bins. In particular, it uses the representatives $x^B$ instead of the exact observations $X_i$ in $\hat{\Phi}_n$, which is the reason for the additional $\lambda^2 Dh$ term in $\xi_n$. Similarly to Algorithm 1, it can be improved to a $\log(M)$ computational complexity with binary search trees as detailed in Appendix B.

**ALGORITHM 2:** Bandit algorithm

---

**input :** $\delta, \lambda, C, D, E, T, \sigma, \kappa$
$\hat{c}_0^- = 0; \hat{r}_0^+ = \infty; n = 0$
**while** $\sum_{i=1}^n S_i + X_i \mathbb{1}(\hat{r}_i^+(X_i) \geq \hat{c}_i^- X_i) < T$ **do**
    $n = n + 1$
    Wait $S_n$ and observe $X_n$
    **if** $\hat{r}_{n-1}^+(X_n) \geq \hat{c}_{n-1}^- x^{B(X_n)}$ **then** accept task and receive reward $Y_n$
    **else** reject task
    For $B = B(X_n)$ compute $\hat{r}_n^B, \hat{r}_n^{B+}, \tilde{r}_n^B$ as described by Equations (4.1) to (4.3)
    Compute $\hat{c}_n$ as the unique root of $\hat{\Phi}_n$ defined in Equation (4.4)
    $\hat{c}_n^- = \hat{c}_n - \xi_{n-1}$ as described by Equation (4.5)
**end**

---

## 5 Simulations

Algorithms 1 and 2 are computationally efficient when using binary search trees as described in Appendix B. This section compares empirically these different algorithms on toy examples, considering affine and concave reward functions. The code used in this section is available at `github.com/eboursier/making_most_of_your_time`.

### 5.1 Affine reward function

We first study the simple case of affine reward function $r(x) = x - 0.5$. The profitability function is increasing and Algorithm 3, given in Appendix A.3, can be used in this specific case. We consider a uniform distribution on $[0, 3]$ for task distribution, a Poisson rate $\lambda = 1$ and a uniform distribution on $[-1, 1]$ for the noise $\varepsilon$. As often in bandit algorithms, we tune the constants scaling the uncertainty terms $\eta$ and $\xi$ for better empirical results.

Figure 1 shows the typical decisions of Algorithms 1 and 2 on a single run of this example. The former especially seems to take almost no suboptimal decision. This is because in simpler settings with good margin conditions, the regret of Algorithm 1 can be shown to be only of order $\log(T)$. On the other hand, it can be seen how Algorithm 2 progressively eliminates bins, represented by red traces.

Figure 2 gives the upper estimate of the reward function by representing $\hat{r}_n^{B+}/x^B$ for each bin and the lower estimate of $\hat{c}_n^-$ after a time $t = 10^5$. It illustrates how the algorithm takes decisions at this stage. Especially, the bins with estimated profitability above $\hat{c}_n^-$, while their actual profitability is below $c^*$, still have to be eliminated by the algorithm. Some bins are badly estimated, since the algorithm stops accepting and estimating them as soon as they are detected suboptimal. The algorithm thus adapts nicely to the problem difficulty, by eliminating very bad bins way faster than slightly suboptimal ones.

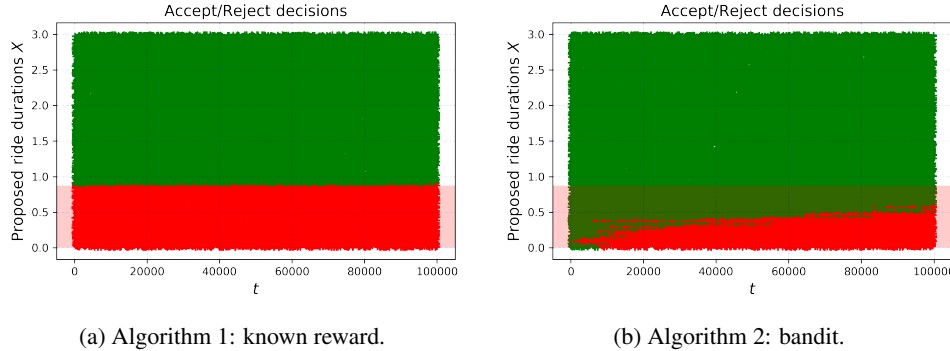

(a) Algorithm 1: known reward.        (b) Algorithm 2: bandit.

Figure 1: Accept/reject decisions on a single run. A single point corresponds to a proposed task, with the time of appearance on $x$ axis and task durations on $y$ axis. It is marked in green (resp. red) if it is accepted (resp. rejected) by the algorithm, while the light red area highlights the suboptimal region. Each task in this colored region thus has a profitability below $c^*$.

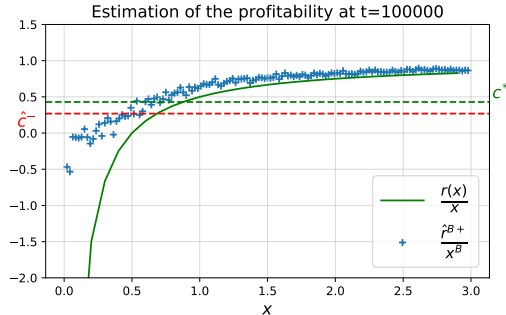

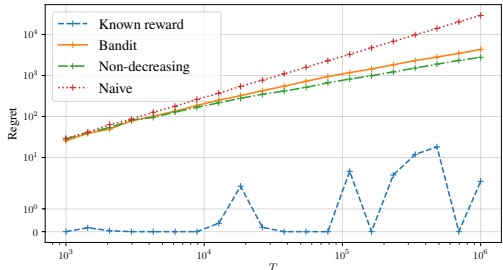

Figure 2: Estimations of $r(\cdot)$ and $c^*$ by Algorithm 2 at $t = 10^5$.

Figure 3: Evolution of regret with $T$ for different algorithms.

Figure 3 shows in logarithmic scale the evolution of regret with the horizon $T$ for the different algorithms. The regret is averaged over 50 runs (and 500 runs for Known reward). Algorithm 1 performs way better than its $\sqrt{T}$ bound, due to a better performance with margin conditions as mentioned above. Its regret is non-monotonic in $T$ here and is sometimes negative, in which case we round it to 0 for clarity. This is only due to the variance in the Monte-Carlo estimates of the expected regret. For such a small regret, much more than 500 runs are required to observe accurately the expected value. We unfortunately could not afford to simulate more runs for computational reasons.

*Non-decreasing* corresponds to Algorithm 3 and performs slightly better than Algorithm 2 here. This remains a small improvement since Algorithm 2 benefits from margin conditions as shown in Appendix A.2. Moreover, this improvement comes with additional computations. Clearly, all these algorithms perform way better than the naive algorithm, which simply accepts every task. We indeed observe that learning occurs and the regret scales sublinearly with $T$ for all non-naive algorithms.

## 5.2 Concave reward function

This section now considers the case of the concave reward function $r(x) = -0.3x^2 + x - 0.2$. The profitability is not monotone here, making Algorithm 3 unadapted. We consider the same task distribution as in Section 5.1 and a Gaussian noise of variance $0.1$ for $\varepsilon$. Similarly to the affine case, Figures 4 to 6 show the behaviors of the different algorithms, as well as the regret evolution. Here as well, the regret is averaged over 50 runs (and 500 runs for Known reward).

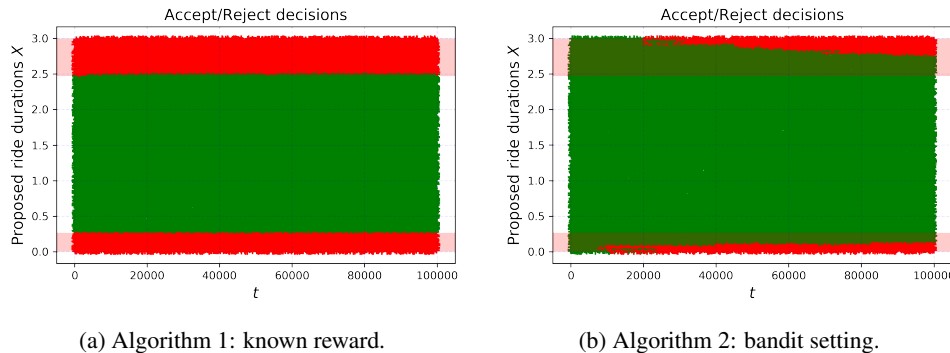

(a) Algorithm 1: known reward.

(b) Algorithm 2: bandit setting.

Figure 4: Accept/Reject decisions on a single run, illustrated as in Figure 1.

The problem is now harder than in the previous affine case, since there are multiple suboptimal regions. Algorithm 1 still performs very well for the same reasons, while Algorithm 2 requires more time to detect suboptimal tasks, but still performs well in the end. Similarly to Figure 3, the regret of Known reward is non-monotonic at first here. However it is increasing for larger values of $T$, as the variance becomes negligible with respect to the regret in this setting.

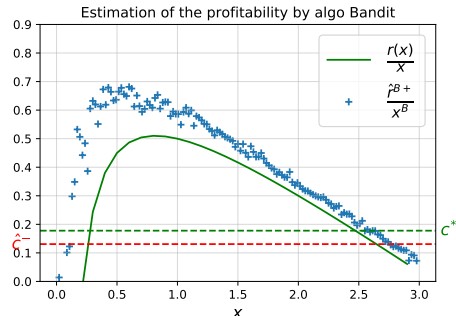

Figure 5: Estimation of profitability function by Algorithm 2 after a time $t = 10^5$.

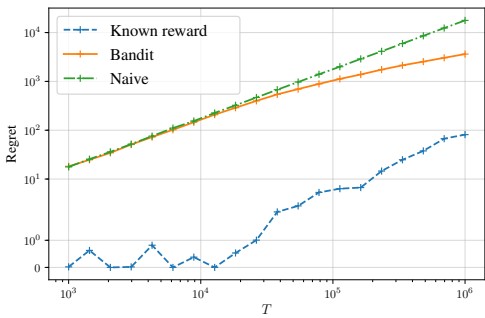

Figure 6: Evolution of regret with time $T$.

## 6 Conclusion

We introduced the problem of online allocation of time where an agent observes the duration of a task before accepting/declining it. Its main novelty and difficulty resides in the estimation and computation of a threshold, which depends both on the whole duration distribution and the reward function. After characterizing a baseline policy, we proposed computationally efficient learning algorithms with optimal regret guarantees, when the reward function is either known or unknown. Even faster learning rates are shown in Appendix A under stronger assumptions on the duration distribution or the reward function.

The model proposed in this work is rather simple and aims at laying the foundations for possible extensions, that might be better adapted to specific applications. We believe that Algorithm 2 can be easily adapted to many natural extensions. For instance, we here assume that the reward function only depends on the duration of the task, while it could also depend on additional covariates; using contextual bandits techniques with our algorithm should directly lead to an optimal solution in this case. Similarly, time is the only limited resource here, while extending this work to multiple limited resources seems possible using knapsack bandits techniques. It is also possible to extend it to several actions (instead of a single accept/decline action) by simultaneously estimating multiple rewards functions.

On the other hand, some extensions deserve careful considerations, as they lead to complex problems. In call centers for example, the agent might be able to complete several tasks simultaneously instead of a single one and characterizing the benchmark becomes much more intricate in this setting. Also, we consider stochastic durations/rewards here. Extending this work to the adversarial case is far from obvious. We yet believe this might be related to online knapsack problems and the use of a competitive analysis might be necessary in this case. For ride-hailing applications, the spatial aspect is also crucial as a driver moves from a starting point to his destination during each ride. Adding a state variable to the model might thus be interesting here.

Finally, we believe this work only aims at proposing a first, general and simple model for online learning problems of time allocation, which can be extended to many settings, depending on the considered application.

### Acknowledgements

E. Boursier was supported by an AMX scholarship. V. Perchet acknowledges support from the French National Research Agency (ANR) under grant number #ANR-19-CE23-0026 as well as the support grant, as well as from the grant "Investissements d'Avenir" (LabEx Ecodec/ANR-11-LABX-0047). This research project received partial support from the COST action GAMENET. M. Scarsini is a member of INdAM-GNAMPA. His work was partially supported by the INdAM-GNAMPA 2020 Project "Random walks on random games" and the Italian MIUR PRIN 2017 Project ALGADIMAR "Algorithms, Games, and Digital Markets." He gratefully acknowledges the kind hospitality of Ecole Normale Supérieure Paris-Saclay, where this project started.

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
