## List of symbols

$a_i$      decision concerning $i$-th task

$B_j$      $j$-th bin

$c^*$      unique root of the function $\Phi$

$c_n$      unique root of $\Phi_n$

$\hat{c}_n$      estimate of $c^*$

$\hat{c}_n^-$      lower estimate of $c^*$

$\tilde{c}$      profitability threshold associated to $r_{\mathcal{E}}$

$C$      upper bound for $X_i$

$D$      upper bound for $r$

$E$      lower bound for $r$

$\mathcal{E}$      subset of sub-profitable tasks, introduced in Lemma 4.3

$h$      size of bins

$\mathcal{H}_t$      history at time $t$

$K$      $\mathrm{card}(\mathrm{supp}(X_1))$

$K_n$      $\mathrm{card}\{X_i \mid 1 \le i \le n\}$

$L$      constant of the Hölder class, defined in Definition 4.1

$M$      number of bins

$n_B$      number of received task proposals in the bin $B$

$N_B$      $\sum_{i=1}^{n} \mathbb{1}(X_i \in B \text{ and } a_i = 1)$

$p$      reward per time unit function, defined in Equation (A.1)

$p_n$      empirical counterpart of $p$, defined in Equation (A.2)

$q_\varepsilon$      $\mathbb{E}[X \mathbb{1}(r(X) \ge (c^* + \varepsilon)X)]$, defined in Equation (3.2)

$r$      reward function

$\hat{r}_n$      estimate of $r$

$r(X_i)$      $\mathbb{E}[Y_i]$

$\hat{r}_n^B$      $\hat{r}_n(x) \; \forall x \in B$, defined in Equation (4.1)

$\hat{r}_n^+$      upper estimate of $r$

$\tilde{r}_n^B$      defined in Equation (4.3)

$r_{\mathcal{E}}(x)$      $r(x) \mathbb{1}(x \notin \mathcal{E})$, introduced in Lemma 4.3

$R$      regret, defined in Equation (2.5)

$s$      threshold

$s^*$      threshold such that $r(s^*)/s^* = c^*$

$s_n$      $\min \mathcal{S}_n$

$S_i$      idling time after $i$-th task

$\mathcal{S}_n$      set of all potentially optimal thresholds, defined in Equation (A.4)

$T$      problem horizon

$t_n$      time at which the $n$-th task is proposed

$\mathcal{T}_n$      amount of time after $n$ proposals

$U_\pi$      expected reward, defined in Equation (2.1)

$v$      value function

$w$      bound on the value function

$x^B$      $\min\{x \in B\}$

$X_i$      duration of the $i$-th task

$Y_i$      reward of the $i$-th task

$\alpha$      parameter of the margin condition

$\beta$      exponent of the Hölder class, defined in Definition 4.1

$\gamma_n$      $\dfrac{\lambda \mathbb{E}[r(X_n) \mathbb{1}(r(X_n) \ge c_n X_n)]}{1 + \lambda \mathbb{E}[X_n \mathbb{1}(r(X_n) \ge c_n X_n)]}$, defined in Equation (3.4)

$\delta$      probability bound, introduced in Proposition 3.1

$\Delta_{\min}$      $\inf\{c^* x - r(x) \mid r(x) < c^* x\}$, introduced in Theorem A.1

$\varepsilon_i$      subgaussian noise

$\zeta_n$      defined in Equation (A.3)

$\eta_{n-1}$      defined in Equation (4.2)

$\theta$      total number of accepted task proposals

$\kappa$      universal constant

$\lambda$      arrival rate of Poisson process

$\xi_{n-1}$      defined in Equation (4.5)

| | |
|---|---|
| $\pi$ | policy |
| $\pi^*$ | threshold policy |
| $\sigma^2$ | proxy of the subgaussian noise |
| $\Phi$ | function $c \mapsto \lambda \mathbb{E}\left[(r(X) - cX)_+\right] - c$, defined in Equation (2.4) |
| $\Phi_n$ | empirical counterpart of $\Phi$, defined in Equation (3.1) |

# Appendix

# A   Fast rates

In this section, we investigate several cases where the general slow rate convergence can be improved. We use two assumptions about the distribution of task durations $X_i$ and one about the reward function $r$. For the sake of clarity, the proofs of all the results from this section are deferred to Appendix F.

## A.1   Finite support

The first assumption that can be used to obtain fast rates of convergence is the so-called "finite support assumption". Under this assumption, the learning algorithm is quite simple and it consists in keeping an under-estimation $\hat{c}_n^-$ of $c^*$ and an upper-estimation $\hat{r}_n^+(x)$ of $r(x)$ for all different values of $x$, whose number is denoted by $K = \mathrm{card}(\mathrm{supp}(X_1))$.

The finite algorithm is then based on Algorithm 2, with the difference that each bin corresponds to a single value of the support of the distribution: $B_j = \{x_j\}$, where $x_j$ is the $j$-th element in $\mathrm{supp}(X_1)$. The estimates $\hat{c}_n$ and $\hat{r}_n$ are then computed similarly to Algorithm 2, but their uncertainty bounds are tighter. This yields the following optimistic estimates:

$$\hat{r}_n^+(x) = \hat{r}_n(x) + \sigma\sqrt{\frac{\ln(K/\delta)}{2N_B}} \qquad \text{for } B = \{x\} \text{ and}$$

$$\hat{c}_n^- = \hat{c}_n - 2\lambda\sqrt{\sigma^2 + \frac{(D-E)^2}{4}}\sqrt{\frac{\ln(1/\delta)}{n}} - \lambda\sigma\sqrt{\frac{K}{2n}} - 8\lambda\frac{K(D-E)}{n}.$$

The task $X_{n+1}$ is then accepted if and only if $\hat{r}_n^+(X_{n+1}) \geq \hat{c}_n^- X_{n+1}$.

**Theorem A.1.** *Assume that the support of $X$ has cardinality $K$. Then, taking $\delta = 1/T$, the finite algorithm's regret scales as*

$$R(T) \leq \kappa_2(1 + \lambda C)\max(\sigma, D - E)\sqrt{KT\ln(T)},$$

*where $\kappa_2$ is some universal constant independent of any problem parameter.*

*Moreover, if $\Delta_{\min} := \inf\{c^*x - r(x) \mid r(x) < c^*x\}$, then*

$$R(T) \leq \kappa_3(1 + \lambda^2 C^2)\max(\sigma, D - E)^2 K \frac{\ln T}{\Delta_{\min}},$$

*for another universal constant $\kappa_3$.*

*Remark* A.2. The algorithm complexity scales with $K$ in this particular setting. The size $K$ of the support is assumed to be known for simplicity. For an unknown $K$, this algorithm can be adapted by using the observed value $K_n := \mathrm{card}\{X_i \mid 1 \leq i \leq n\}$ instead of $K$ and restarting every time $K_n$ increases. Such an algorithm yields a similar performance, if not better in practice.

## A.2   Margin condition

The finite support assumption gives very fast learning rates but is quite strong. It is possible to study some "intermediate" regime between the general Hölder case and the finite support assumption, thanks to the margin condition recalled below. Intuitively, this condition states that slightly suboptimal tasks have a small probability of being proposed, thus limiting the regret from declining them.

**Definition A.3.** The distribution of task durations $X_i$ satisfies the *margin condition* with parameter $\alpha \geq 0$ if there exists $\kappa_0 > 0$ such that for every $\varepsilon > 0$,

$$\mathbb{P}\left(c^*X - \varepsilon \leq r(X) < c^*X\right) \leq \kappa_0\varepsilon^\alpha.$$

If the margin condition $\alpha$ is small enough (i.e., the problem is not too easy and $\alpha < 1$), then we can apply the very same algorithm as in the general case when $r$ is $(L, \beta)$-Hölder. We also assume that the distribution of task durations has a positive density lower-bounded by $\underline{\kappa} > 0$ and upper bounded by $\overline{\kappa} > 0$; with a slight abuse of notation, we call this being Lebesgue-equivalent.

**Theorem A.4.** *If $r$ is $(L, \beta)$-Hölder, the distribution of $X$ satisfies the margin condition with (unknown) parameter $\alpha \in [0, 1)$, and is Lebesgue-equivalent, then the regret of Algorithm 2 (with same $\delta$ and $M$ as in Theorem 4.5) scales as*

$$R(T) \leq \kappa_4 \sigma^{1+\alpha} (\lambda C)^{1+\alpha} L^{\frac{1+\alpha}{2\beta+1}} (\lambda T + 1)^{1 - \frac{\beta}{2\beta+1}(1+\alpha)} \log(\lambda T + 1)^{\frac{1+\alpha}{2}},$$

*where $\kappa_4$ is a constant depending only on $\underline{\kappa}$ and $\kappa_0$ defined in the margin condition assumption.*

*Remark* A.5. The proof follows the lines of Theorem 4.1 in Perchet and Rigollet (2013). The term $\log(\lambda T + 1)$ can be removed with a refined analysis using decreasing confidence levels $\delta_n = 1/n^2$ instead of a fixed $\delta$.

For larger margin conditions $\alpha \geq 1$, improved rates are still possible using an adaptive binning instead. The analysis should also follow the lines of Perchet and Rigollet (2013).

## A.3   Monotone profitability function

We consider in this section an additional assumption on the profitability function $x \mapsto r(x)/x$, motivated by practical considerations. We assume that this function is non-decreasing (as the impact of some fixed cost declines with the task duration) – the analysis would follow the same line for a non-increasing profitability function (to model diminishing returns for instance). A sufficient condition is convexity of the reward function $r$ with $r(0) \leq 0$. It is not difficult to see that a non-decreasing profitability function ensures the existence of a threshold[1] $s^* \in [0, C]$, such that $r(s^*)/s^* = c^*$. Moreover, by monotonicity, it is optimal to accept a task with duration $x$ if and only if $x \geq s^*$. Hence the problem reduces to learning this optimal threshold based on the reward gathered in the first tasks. In particular, we consider plug-in strategies that accept any task whose duration is large enough (and decline the other ones). The reward per time unit of these strategies is then a function of the threshold $s$, defined by

$$p(s) = \frac{\lambda \, \mathbb{E}[r(X) \, \mathbb{1}(X \geq s)]}{1 + \lambda \, \mathbb{E}[X \, \mathbb{1}(X \geq s)]}. \tag{A.1}$$

Our policy uses and updates a threshold $s_n$ for all rounds. Let $S = 2(\lambda T + 1)$ be an upper bound on the total number of received tasks with high probability. As the objective is to find the optimum of the function $p(\cdot)$, we introduce its empirical counterparts defined, at the end of stage $n$, by

$$p_n(s) = \frac{\frac{\lambda}{n} \sum_{i=1}^{n} Y_i \, \mathbb{1}(X_i \geq s)}{1 + \frac{\lambda}{n} \sum_{i=1}^{n} X_i \, \mathbb{1}(X_i \geq s)}. \tag{A.2}$$

Given some confidence level $\delta \in (0, 1]$ – to be chosen later –, we also introduce the error term

$$\zeta_n = \left( \sqrt{\sigma^2 + \frac{(D - E)^2}{4}} + \frac{D - E}{\sqrt{2}} (\lambda C + 2) \right) \sqrt{\frac{\ln\left(\frac{2(S+1)}{\delta}\right)}{n - 1}} + \lambda \frac{D - E}{n}, \tag{A.3}$$

in the estimation of $p(\cdot)$ by $p_n(\cdot)$, see Lemma F.1 in the Appendix.

The policy starts with a first threshold $s_1 = 0$. After observing a task, the first estimate $p_1(\cdot)$ is computed as in Equation (A.2) and we proceed inductively. After $n - 1$ tasks, given the estimate $p_{n-1}(\cdot)$ on $[s_{n-1}, C]$, we compute a lower estimate of $p(s^*)$ by $\max_s p_n(s) - \zeta_n$ (for notational convenience, we also introduce $s_{n-1}^* = \arg\max_{s \in [s_{n-1}, C]} p_{n-1}(s)$). A threshold is then potentially optimal if its associated reward per time unit (plus the error term) $p_n(s) + \zeta_n$ exceeds this lower estimate. Mathematically, the set of all potentially optimal thresholds is therefore:

$$\mathcal{S}_n = \left\{ s \in [s_{n-1}, C] \, \middle| \, (p_{n-1}(s) - p_{n-1}(s_{n-1}^*)) \left( \frac{1}{\lambda} + \frac{1}{n-1} \sum_{i=1}^{n-1} X_i \, \mathbb{1}(X_i \geq s) \right) + 2\zeta_{n-1} \geq 0 \right\}. \tag{A.4}$$

---

[1] For non-continuous reward, we only have that $\lim_{\substack{x \to s^* \\ x < s^*}} \frac{r(x)}{x} \leq c^* \leq \lim_{\substack{x \to s^* \\ x > s^*}} \frac{r(x)}{x}$, but the following remains valid.

We then define the new threshold to be used in the next stage as $s_n = \min \mathcal{S}_n$. The rationale behind this choice is that it provides information for all the (potentially optimal) thresholds in $\mathcal{S}_n$. The pseudo-code is given in Algorithm 3 below.

We can finally state the main result when profitability is monotone. Regret scales as $\sqrt{T}$ independently of the regularity of the reward function $r$.

**Theorem A.6.** *Assume the profitability function $x \mapsto r(x)/x$ is non-decreasing. Then, the regret of Algorithm 3 (taking $\delta = 1/T^2$) scales as*

$$R(T) \leq \kappa_5 (\sigma + D - E)(1 + \lambda C)\sqrt{(\lambda T + 1)\ln(\lambda T)}, \tag{A.5}$$

*where $\kappa_5$ is a constant independent from any problem parameter.*

---

**ALGORITHM 3:** Non-decreasing profitability algorithm

---

**input :** $\delta, \lambda, C, D, \sigma, E, T$
$s_1 = 0; S = 2\lambda T + 1; n = 0$
**while** $\sum_{i=1}^{n} S_i + X_i \mathbb{1}(X_i \geq s_i) < T$ **do**

> $n = n + 1$
> Wait $S_n$ and observe $X_n$
> **if** $X_n \geq s_n$ **then** accept task and receive reward $Y_n$
> **else** reject task
>
> Compute for $s \in [s_n, C]$, $\qquad p_n(s) = \frac{\lambda \frac{1}{n}\sum_{i=1}^{n} Y_i \mathbb{1}(X_i \geq s)}{1 + \lambda \frac{1}{n}\sum_{i=1}^{n} X_i \mathbb{1}(X_i \geq s)}$
> Compute $s_n^* = \arg\max_{s \in [s_n, C]} p_n(s)$
> Let $\mathcal{S}_{n+1} = \left\{ s \in [s_n, C] \,|\, (p_n(s) - p_n(s_n^*))\left(\frac{1}{\lambda} + \frac{1}{n}\sum_{i=1}^{n} X_i \mathbb{1}(X_i \geq s)\right) + 2\zeta_n \geq 0 \right\}$
> Set $s_{n+1} = \min \mathcal{S}_{n+1}$

**end**

---

*Remark* A.7. The function $p_n(s)$ is piecewise constant in Algorithm 3 and thus only requires to be computed at all the points $X_i$. The algorithm then requires to store the complete history of tasks and has a complexity of order $n$ when receiving the $n$-th proposition.
However, a trick similar to Remark 3.5 leads to improved complexity. The algorithm actually requires to compute $p_n(s)$ only for all $s \in \mathcal{S}_n$. Only tasks of length in $\mathcal{S}_n$ then have to be individually stored and the computation of all "interesting" values of $p_n$ scales with the number of $X_i$ in $\mathcal{S}_n$. This yields a sublinear complexity in $n$ under regularity conditions on the distribution of $X$.

## A.4 Lower bounds

All the algorithms we propose are actually "optimal" for their respective class of problems. Here, optimality must be understood in the minimax sense, as a function of the horizon $T$ and up to $\text{poly}\log(T)$ terms. More formally, a minimax lower-bound for a given class of problems (say, $(L, \beta)$-Hölder reward functions with bandit noisy feedback) states that, for any given algorithm, there always exists a specific problem instance such that $R(T) \geq \kappa' T^\gamma$ for some number $\gamma \in [0, 1]$ and $\kappa'$ is independent of $T$. An algorithm whose regret never increases faster than $\kappa'' T^\gamma \log^{\gamma'}(T)$, for some constants $\kappa''$ and $\gamma'$ independent of $T$ is then *minimax optimal*.

**Proposition A.8.** *Our algorithms are all minimax optimal for their respective class of problems.*

The proof is rather immediate and postponed to Appendix F.3. Indeed, the agent is facing a problem more complicated than a contextual (one-armed) bandit problem because $c^*$ is unknown at first. On the other hand, if we assume that it is given to the agent, the problem is actually a one-armed contextual bandit problem. As a consequence, lower bounds for this problem are also lower-bounds for the agent's problem; as lower and upper bounds match (up to $\text{poly}\log$ terms), the devised algorithms are optimal.

## B  Fast implementation using search trees

First recall that the complexity of computing the $n$-th decision of Algorithm 1 is of order $n$ in worst cases, while Algorithm 2 scales with $M$, which is of order $T^{\frac{1}{2\beta+1}}$. The computational complexities of these algorithms at the $n$-th round can actually be respectively improved to $\log(n)$ and $\log(M)$,

while keeping the same space complexity, by the use of augmented balanced binary search trees data structures, e.g., augmented red black trees (Cormen et al., 2009, Chapters 12-14). Such kind of trees stores nodes with keys in an increasing order. The search, insert and deletion function can thus all be computed in a time $\log(n)$, where $n$ is the number of nodes in the tree. An augmented structure allows to store additional information at each node, which is here used to compute the root of $\Phi_n$.

**Fast Algorithm 1.**  For Algorithm 1, each task $i$ is stored as a node with the key $r(X_i)/X_i$, with the additional piece of information $(r(X_i), X_i)$. Using augmented trees, it is possible to also store in each node $i$ the sum of $r(X_j)$ and $X_j$ for all the nodes $j$ in its right subtree, and similarly for its left subtree with the same complexity.

The main step of Algorithm 1 is the computation of $c_n$, which can now be computed in a time $\log(n)$, thanks to the described structure. Indeed, note that

$$\Phi_n \left( \frac{r(X_i)}{X_i} \right) = \frac{\lambda}{n} \left( \sum_{j: \frac{r(X_j)}{X_j} > \frac{r(X_i)}{X_i}} r(X_j) - \frac{r(X_i)}{X_i} \sum_{j: \frac{r(X_j)}{X_j} > \frac{r(X_i)}{X_i}} X_j \right) - \frac{r(X_i)}{X_i}.$$

As the right subtree of the root exactly corresponds to the tasks $j$ with a larger profitability than the root, $\Phi_n(c)$ can be computed in time 1 where $c$ is the profitability of the root node, using the additional stored information. As $\Phi_n$ is decreasing, the algorithm continues in the left subtree if $\Phi_n(c) < 0$ and in the right subtree otherwise. Evaluating $\Phi_n$ at the key of the following node also takes a time 1 using the different stored information. The algorithm then searches the tree in a time $\log(n)$ to find the task $i^*$ with the largest profitability such that $\Phi_n \left( \frac{r(X_{i^*})}{X_{i^*}} \right) > 0$.

Note that $\Phi_n$ is piecewise linear with a partition given by the profitability of the received tasks. In particular, it is linear on $[\frac{r(X_{i^*})}{X_{i^*}}, c_n]$ and $c_n$ is given by

$$c_n = \frac{\lambda \sum_{j: \frac{r(X_j)}{X_j} > \frac{r(X_{i^*})}{X_{i^*}}} r(X_j)}{n + \lambda \sum_{j: \frac{r(X_j)}{X_j} > \frac{r(X_{i^*})}{X_{i^*}}} X_j}.$$

Recall that it was suggested in Remark 3.5 to remove all tasks with profitability outside

$$\left[ c_n - \lambda(D - E)\sqrt{\frac{\ln(1/\delta)}{2n}}, \ c_n + \lambda(D - E)\sqrt{\frac{\ln(1/\delta)}{2n}} \right]$$

and to only store the sum of rewards and durations for those above $c_n + \lambda(D-E)\sqrt{\frac{\ln(1/\delta)}{2n}}$. Removing all nodes above or below some threshold in a search tree is called the split operation and can be done in time $\log(n)$ with the structures used here. Thanks to this, keeping a reduced space complexity is possible without additional computational cost.

**Fast Algorithm 2.**  For Algorithm 2, each node represents a bin with the key $(\frac{\tilde{r}_n^B}{x^B}, B)$ and the additional information $(n_B \tilde{r}_n^B, n_B x^B)$. $B$ is also required in the key to handle the case of bins with same profitabilities.

Here again at each node, we also store the sum of the additional information of the left and right subtrees. Note that at each time step, the algorithm updates $\tilde{r}_n^B$ for the bin of the received task. This is done by deleting the node with the old value[2] of $\tilde{r}_n^B / x^B$ and insert a node with the new value.

The computation of $\hat{c}_n$ then follows the lines of the computation of $c_n$ in Algorithm 1 above.

Unfortunately, search trees do not seem adapted to Algorithm 3 since maximizing the function $p_n(s)$ requires to browse the whole tree here.

---

[2] The algorithm also requires to store a list to find $\tilde{r}_n^B / x^B$ in time 1 for any bin $B$.

# C  Proofs of Section 2

*Proof of Proposition 2.1.* For $h > 0$, let $B(t, h)$ denote the event "the agent is proposed exactly one task in the interval $[t, t+h]$." Since the probability of receiving more than one task in this interval is $o(h)$, one has for $t < T$

$$v(t) = \mathbb{P}(B(t,h))\,\mathbb{E}[\max(r(X) + v(t+X), v(t+h)) \mid B(t,h)] + (1 - \mathbb{P}(B(t,h)))v(t+h) + o(h)$$
$$= (\lambda h + o(h))\,\mathbb{E}[\max(r(X) + v(t+X), v(t+h)) \mid B(t,h)] + (1 - \lambda h + o(h))v(t+h) + o(h).$$

By definition of the value function, the optimal strategy accepts a task $X$ at time $t$ if and only if $r(X) + v(t+X) \geq v(t)$. Hence

$$\frac{v(t+h) - v(t)}{h} = -(\lambda + o(1))\,\mathbb{E}[(r(X) + v(t+X) - v(t+h))_+ \mid B(t,h)] + o(1). \qquad \text{(C.1)}$$

Letting $h \to 0$ one has $v'(t) = -\lambda\,\mathbb{E}[(r(X) + v(t+X) - v(t))_+]$.

Since $(t, v) \mapsto -\lambda\,\mathbb{E}[(r(X) + v(t+X) - v(t))_+]$ is uniformly Lipschitz-continuous in $v$, uniqueness of the solution follows the same lines as the proof of the Picard-Lindelöf (a.k.a. Cauchy-Lipschitz) theorem. Hence, $v$ is the unique solution of the dynamic programming Equation (2.2).

Note that $w$ is actually the unique solution of the same equation without the boundary effect

$$\begin{cases} w'(t) = -\lambda\,\mathbb{E}\left[(r(X) + w(t+X) - w(t))_+\right] & \text{for all } t \in \mathbb{R}, \\ w(T) = 0. \end{cases} \qquad \text{(C.2)}$$

Similarly to the argument above, $w$ is the value function of the optimal strategy of an alternative program, defined as the original program, with the difference that, if the agent ends at time $t > T$, then she suffers an additional loss $(t - T)c^*$ in her reward.

The optimal strategy does not only maximize the value function at time $t = 0$, but actually maximizes it for any time $t \in [0, T]$. By definition, any strategy earns less in the alternative program than in the original program. By optimality of the strategy giving the value $v$, this yields $v(t) \geq w(t)$ for any time $t$.

By translation, $w(\cdot - C)$ is also the value of the optimal strategy in the alternative program, but with horizon $T + C$. As the length of any task is at most $C$, following the optimal strategy in the original program of horizon $T$ and rejecting all tasks proposed between $T$ and $T + C$ yields exactly the value $v$ in this *delayed* alternative program; thus $w(t - C) \geq v(t)$ by optimality. $\qquad \square$

*Proof of Theorem 2.2.* Recall that the strategies considered in the proof of Proposition 2.1 accept a task $X$ at time $t$ if and only if $r(X) + v(t+X) \geq v(t)$ where $v$ is its value function. Thus, the optimal strategy in the alternative program described in the proof of Proposition 2.1 accepts a task $X$ if and only if $r(X) \geq c^*X$. Moreover, the cumulative reward of this strategy in the original program is larger than $w(0)$. The relation between $v$ and $w$ given by Proposition 2.1 then yields the result. $\qquad \square$

# D  Proofs of Section 3

*Proof of Proposition 3.1.* Let $\varepsilon > 0$. One has,

$$\begin{aligned} \mathbb{P}(c_n - c^* > \varepsilon) &= \mathbb{P}(\Phi_n(c_n) < \Phi_n(c^* + \varepsilon)) \text{ since } \Phi_n \text{ is decreasing} \\ &= \mathbb{P}(0 < \Phi_n(c^* + \varepsilon)) \text{ since } c_n \text{ is the root of } \Phi_n \\ &\leq \mathbb{P}(\varepsilon < \Phi_n(c^* + \varepsilon) - \Phi(c^* + \varepsilon)) \text{ since } \Phi(c^* + \varepsilon) \leq -\varepsilon \\ &\leq \exp\left(\frac{-2n\varepsilon^2}{\lambda^2(D-E)^2}\right), \end{aligned}$$

where the last inequality follows from Hoeffding's inequality. $\qquad \square$

**Proposition D.1.** *For all $n \geq 1$, the following bound holds*

$$(c^* - \gamma_n)\,\mathbb{E}\left[X_n\,\mathbb{1}(r(X_n) \geq c_n X_n) + S_n\right] \leq \frac{\lambda(D-E)C}{2}\sqrt{\frac{\pi}{2n}}.$$

*Proof of Proposition D.1.* First note that the definition of $c^*$ implies

$$c^* = \frac{\lambda \, \mathbb{E}[r(X) \, \mathbb{1}(r(X) \geq c^* X)]}{1 + \lambda \, \mathbb{E}[X \, \mathbb{1}(r(X) \geq c^* X)]}.$$

Let us denote $\nu$ the numerator and $\mu$ the denominator of the above expression. First decompose

$$\mathbb{1}(r(X) \geq c_n X) = \mathbb{1}(r(X) \geq c^* X) + \mathbb{1}(c^* X > r(X) \geq c_n X) - \mathbb{1}(c_n X > r(X) \geq c^* X).$$

Denote

$$Q = \lambda \, \mathbb{E}\left[ X \Big( \mathbb{1}(c^* X > r(X) \geq c_n X) - \mathbb{1}(c_n X > r(X) \geq c^* X) \Big) \right].$$

One has

$$
\begin{aligned}
\gamma_n &= \frac{\nu + \lambda \, \mathbb{E}[r(X)(\mathbb{1}(c^* X > r(X) \geq c_n X) - \mathbb{1}(c_n X > r(X) \geq c^* X))]}{\mu + \lambda \, \mathbb{E}[X(\mathbb{1}(c^* X > r(X) \geq c_n X) - \mathbb{1}(c_n X > r(X) \geq c^* X))]} \\
&\geq \frac{\nu + \lambda \, \mathbb{E}[c_n X(\mathbb{1}(c^* X > r(X) \geq c_n X) - \mathbb{1}(c_n X > r(X) \geq c^* X))]}{\mu + Q} \\
&= c^* + \frac{\mathbb{E}\left[(c_n - c^*)Q\right]}{\mu + Q} \\
&\geq c^* - \frac{\lambda C \, \mathbb{E}[(c_n - c^*)_+]}{1 + \lambda \, \mathbb{E}\left[X_n \, \mathbb{1}(r(X_n) \geq c_n X_n)\right]} \\
&\geq c^* - \frac{\lambda(D - E)C}{2} \sqrt{\frac{\pi}{2n}} \frac{1}{\mathbb{E}\left[X_n \, \mathbb{1}(r(X_n) \geq c_n X_n) + S_n\right]},
\end{aligned}
$$

where the last inequality follows from Proposition 3.1. $\qquad\square$

*Proof of Theorem 3.3.* From Equation (3.3) one has

$$
\begin{aligned}
R(T) &= \mathbb{E}\left[ \sum_{n=1}^{\theta}(c^* - \gamma_n) \, \mathbb{E}\left[X_n \, \mathbb{1}(r(X_n) \geq c_n X_n) + S_n\right] \right] + c^* \left( T - \mathbb{E}\left[ \sum_{n=1}^{\theta} X_n \, \mathbb{1}(r(X_n) \geq c_n X_n) + S_n \right] \right) \\
&\leq \mathbb{E}\left[ \sum_{n=1}^{\theta}(c^* - \gamma_n) \, \mathbb{E}\left[X_n \, \mathbb{1}(r(X_n) \geq c_n X_n) + S_n\right] \right] \\
&\leq \frac{\lambda(D - E)C}{2} \sqrt{\frac{\pi}{2}} \, \mathbb{E}\left[ \sum_{n=1}^{\theta} \sqrt{\frac{1}{n}} \right] \quad \text{by Proposition D.1} \\
&\leq \lambda(D - E)C \sqrt{\frac{\pi}{2}} \, \mathbb{E}\left[ \sqrt{\theta} \right] \quad \text{by Wald's formula.}
\end{aligned}
$$

It remains to control the expected number of tasks observed $\mathbb{E}\,\theta$. First remark that

$$\theta - 1 \leq \min\{n \geq 1 \mid \sum_{i=1}^{n} S_i > T\} - 1 = \sup\{n \geq 1 \mid \sum_{i=1}^{n} S_i \leq T\} \tag{D.1}$$

and that the law of the right-hand side of Equation (D.1) is Poisson of parameter $\lambda T$. So we get that $\mathbb{E}\,\theta \leq \lambda T + 1$ and thus $\mathbb{E}\,\sqrt{\theta} \leq \sqrt{\lambda T + 1}$ by Jensen's inequality. $\qquad\square$

# E   Proofs of Section 4

*Proof of Lemma 4.2.* It follows from Hoeffding's inequality, and the fact that $r$ is Hölder continuous.
□

*Proof of Lemma 4.3.* The proof of this crucial lemma is actually straightforward, and follows from the fact that $c^* \geq 0$ and thus $(r_{\mathcal{E}}(X) - c^* X)_+ = (r(X) - c^* X)_+$. □

*Proof of Proposition 4.4.* Let us, in the first step, assume that all the first $N$ tasks have been accepted, so that $(X_i, Y_i)$ are i.i.d., and $\mathbb{E}[Y_i|X_i] = r(X_i)$. We define

$$\bar{\Phi}_n : c \mapsto \lambda \, \mathbb{E}[\left(\hat{r}_n(X) - cx^{B(X)}\right)_+] - c,$$

where $B(X)$ is the bin corresponding to $X$. Hence for all $c \geq 0$, $\mathbb{E}[\hat{\Phi}_n(c)] = \bar{\Phi}_n(c)$.

Second, remark that for all $c \geq 0$

$$|\bar{\Phi}_n(c) - \Phi(c)| = \lambda |\, \mathbb{E}[(\hat{r}_n(X) - cx^{B(X)})_+ - (r(X) - cX)_+]| \tag{E.1}$$
$$\leq \lambda \, \mathbb{E}[|\hat{r}_n(X) - r(X)|] + \lambda ch \tag{E.2}$$
$$\leq \lambda \, \mathbb{E}[|\hat{r}_n(X) - r(X)|] + \lambda^2 Dh. \tag{E.3}$$

The last inequality is obtained by noting that $c^* \leq \lambda D$, and thus we only consider $c \leq \lambda D$.

Third, for a fixed $c \geq 0$, $\hat{\Phi}_n(c)$ as a function of $(Y_1, \ldots, Y_n)$, is $\frac{\lambda}{n}$-Lipschitz with respect to each variable. Hence,

$$\mathbb{P}(\hat{c}_n - c^* > \varepsilon) = \mathbb{P}(\hat{\Phi}_n(\hat{c}_n) < \hat{\Phi}_n(c^* + \varepsilon)) \text{ since } \hat{\Phi}_n \text{ is decreasing}$$
$$= \mathbb{P}(0 < \hat{\Phi}_n(c^* + \varepsilon)) \text{ since } \hat{c}_i \text{ is the root of } \hat{\Phi}_i$$
$$= \mathbb{P}(-\bar{\Phi}_n(c^* + \varepsilon) < \hat{\Phi}_n(c^* + \varepsilon) - \bar{\Phi}_n(c^* + \varepsilon))$$
$$\leq \mathbb{P}(-\Phi(c^* + \varepsilon) - \lambda \, \mathbb{E}[|\hat{r}_n(X) - r(X)|] - \lambda^2 Dh < \hat{\Phi}_n(c^* + \varepsilon) - \bar{\Phi}_n(c^* + \varepsilon)) \text{ by } Equation \text{ (E.3)}$$
$$\leq \mathbb{P}(\varepsilon - \lambda \, \mathbb{E}[|\hat{r}_n(X) - r(X)|] - \lambda^2 Dh < \hat{\Phi}_n(c^* + \varepsilon) - \bar{\Phi}_n(c^* + \varepsilon)) \text{ since } \Phi(c^* + \varepsilon) \leq -\varepsilon$$
$$\leq \exp\left(\frac{-n(\varepsilon - \lambda \, \mathbb{E}[|\hat{r}_n(X) - r(X)|] - \lambda^2 Dh)^2}{4\lambda^2(\sigma^2 + \frac{(D-E)^2}{4})}\right) \text{ by McDiarmid's inequality.}$$

The last inequality uses McDiarmid's inequality for subgaussian variables (Kontorovich, 2014, Theorem 1). We conclude using Györfi et al. (2002, Corollary 11.2), which yields that $\mathbb{E}[|\hat{r}_n(X) - r(X)|] \leq \kappa \max(\sigma, \frac{D-E}{2})\sqrt{\frac{\log(n)+1}{hn}} + \sqrt{8}\frac{Lh^\beta}{2^\beta}$.

Now, we focus on the case where some of the first $N$ tasks have been declined by the agent. We are going to prove the result by induction. Consider the event where, on the first $n-1$ tasks, it always happened that $\hat{r}_i^+(\cdot) \geq r(\cdot)$ and $\hat{c}_i^- \leq c^*$. As a consequence, on this event, only sub-optimal tasks are declined. Using Lemma 4.3, this yields that the optimal value per time step associated to $r(\cdot)$ and to $r(\cdot)\mathbb{1}(\tilde{r}_n(\cdot) > 0)$ are equal.

As a consequence, the precedent arguments can be applied to the following function

$$\tilde{\Phi}_n : c \mapsto \lambda \, \mathbb{E}\left[(\tilde{r}_n(X) - cx^{B(X)})\right]$$

instead of $\bar{\Phi}_n$. Although Györfi et al. (2002, Corollary 11.2) requires the function $r$ to be $(\beta, L)$-Hölder, it also holds here as $r(\cdot)\mathbb{1}(\tilde{r}_n(\cdot) > 0)$ is $(\beta, L)$-Hölder on every interval of the subdivision $B_1, \ldots, B_M$.

□

Note that in the proof above, we always consider $\varepsilon > \sqrt{8}\lambda\frac{Lh^\beta}{2^\beta} + \lambda^2 Dh$. Thus, by removing the term $\sqrt{8}\lambda\frac{Lh^\beta}{2^\beta} + \lambda^2 Dh$ in $\xi_{n-1}$, all tasks with profitability above $c^* + \varepsilon$ remain observed. With a slight modification of the last argument, it can still be shown that $\hat{\Phi}_n(\hat{c}_n^-) < 0$ with high probability. While this has no influence on the theoretical order of the regret, it is used in the experiments as it yields a significant practical improvement.

*Proof of Theorem 4.5.* Recall that upon the $n$-th call, the agent computes $\hat{r}_{n-1}$ and $\hat{c}_{n-1}$ and accepts the task $X_n \in B$ if and only if $\hat{r}_{n-1}^+(X_n) \geq \hat{c}_{n-1}^- x^{B(X_n)}$. Let us denote by $A(n)$ this event so that the regret can be decomposed into

$$R(T) = c^* T - \mathbb{E}\left[\sum_{n=1}^{\theta} r(X_n)\,\mathbb{1}(A(n))\right]$$

where

$$\theta := \min\{N \in \mathbb{N} \mid \sum_{n=1}^{N} S_n + X_n\,\mathbb{1}\,(A(n)) > T\}.$$

The regret can then be rewritten as

$$R(T) = c^* T - \mathbb{E}\left[\sum_{n=1}^{\theta} \mathbb{E}\left[r(X_n)\,\mathbb{1}(A(n))\right]\right] \quad \text{by Wald's equation}$$

$$= c^* T - \mathbb{E}\left[\sum_{n=1}^{\theta} \hat{\gamma}_n\,\mathbb{E}\left[X_n\,\mathbb{1}(A(n)) + S_n\right]\right],$$

where the reward per time unit of task $n$ is

$$\hat{\gamma}_n := \frac{\lambda\,\mathbb{E}[r(X_n)\,\mathbb{1}(A(n))]}{1 + \lambda\,\mathbb{E}[X_n\,\mathbb{1}(A(n))]}.$$

We further decompose the regret into

$$R(T) = \mathbb{E}\left[\sum_{n=1}^{\theta} (c^* - \hat{\gamma}_n)\,\mathbb{E}\left[X_n\,\mathbb{1}(A(n)) + S_n\right]\right] + c^*\left(T - \mathbb{E}\left[\sum_{n=1}^{\theta} X_n\,\mathbb{1}(A(n)) + S_n\right]\right)$$

$$\leq \mathbb{E}\left[\sum_{n=1}^{\theta} (c^* - \hat{\gamma}_n)\,\mathbb{E}\left[X_n\,\mathbb{1}(A(n)) + S_n\right]\right]$$

$$\leq \mathbb{E}\left[\sum_{n=1}^{\theta} 2C\xi_{n-1} + 4nD\delta + c^* h + 2\,\mathbb{E}[\eta_{n-1}(X_n)]\right],$$

where the last inequality is a consequence of the following Proposition E.1. As a consequence, it only remains to bound the last term

$$\mathbb{E}\left[\sum_{n=1}^{\theta} \mathbb{E}[\eta_{n-1}(X_n)]\right]$$

$$= \mathbb{E}\left[\sum_{n=1}^{\theta} \mathbb{E}\left[\sum_{j=1}^{M} \eta_{n-1}(B_j)\,\mathbb{1}(X_n \in B_j)\right]\right]$$

$$= \mathbb{E}\left[\sum_{n=1}^{\theta} \mathbb{E}\left[\sum_{j=1}^{M} \sqrt{\sigma^2 + \frac{L^2}{4}\left(\frac{C}{M}\right)^{2\beta}}\sqrt{\frac{\ln(M/\delta)}{2N_{B_j}(n-1)}}\,\mathbb{1}(X_n \in B_j)\right] + \theta L\left(\frac{C}{M}\right)^{\beta}\right]$$

$$= \sqrt{\sigma^2 + \frac{L^2}{4}\left(\frac{C}{M}\right)^{2\beta}}\sqrt{\ln(M/\delta)}\,\mathbb{E}\left[\sum_{j=1}^{M}\sum_{n=1}^{\theta} \frac{\mathbb{1}(X_n \in B_j)}{\sqrt{2N_{B_j}(n-1)}}\right] + \theta L\left(\frac{C}{M}\right)^{\beta}$$

$$\leq \sqrt{\sigma^2 + \frac{L^2}{4}\left(\frac{C}{M}\right)^{2\beta}}\sqrt{\ln(M/\delta)}\,\mathbb{E}\left[\sum_{j=1}^{M} \sqrt{N_{B_j}(\theta-1)}\right] + \theta L\left(\frac{C}{M}\right)^{\beta}$$

$$\leq \sqrt{\sigma^2 + \frac{L^2}{4}\left(\frac{C}{M}\right)^{2\beta}}\sqrt{\ln(M/\delta)}\,\mathbb{E}\left[\sqrt{M\theta}\right] + \theta L\left(\frac{C}{M}\right)^{\beta}$$

$$\leq \sqrt{\sigma^2 + \frac{L^2}{4}\left(\frac{C}{M}\right)^{2\beta}}\sqrt{\ln(M/\delta)}\sqrt{M(\lambda T + 1)} + (\lambda T + 1)L\left(\frac{C}{M}\right)^{\beta}$$

Hence, putting all things together, we get

$$R(T) \leq 2\lambda C\Big(4\sqrt{\sigma^2 + \frac{(D-E)^2}{4}}\sqrt{\ln(1/\delta)}\sqrt{\lambda T + 1} + \kappa \max(\sigma, \frac{D-E}{2})\sqrt{\frac{M}{C}\log(e(\lambda T + 1))}\sqrt{\lambda T + 1}$$

$$+ 2\sqrt{2}\frac{LC^{\beta}}{(2M)^{\beta}}(\lambda T + 1) + \lambda D\frac{C}{M}\Big)$$

$$+ 4D(\lambda T + 1)\delta + c^*(\lambda T + 1)\frac{C}{M}$$

$$+ 2\sqrt{\sigma^2 + \frac{L^2}{4}\left(\frac{C}{M}\right)^{2\beta}}\sqrt{\ln(M/\delta)}\sqrt{M(\lambda T + 1)} + 2(\lambda T + 1)L\left(\frac{C}{M}\right)^{\beta}.$$

The result follows from the specific choices $\delta = \frac{1}{T^2}$ and $M = \left\lceil CL^{\frac{2}{2\beta+1}}(\lambda T + 1)^{\frac{1}{2\beta+1}}\right\rceil$. $\qquad\square$

**Proposition E.1.** *For all $n \geq 1$,*

$$0 \leq (c^* - \hat{\gamma}_n)\,\mathbb{E}\left[X_n\,\mathbb{1}(A(n)) + S_n\right] \leq 2C\xi_{n-1} + 4nD\delta + 2\,\mathbb{E}[\eta_{n-1}(X_n)] + c^*h.$$

*Proof.* Recall that

$$c^* = \frac{\lambda\,\mathbb{E}[r(X)\,\mathbb{1}(r(X) \geq c^*X)]}{1 + \lambda\,\mathbb{E}[X\,\mathbb{1}(r(X) \geq c^*X)]} = \frac{\nu}{\mu}.$$

We first decompose the indicator function in the numerator of $\hat{\gamma}_n$. Recall that the task $n$ is accepted (the event $A(n)$ occurs) if $\hat{r}_{n-1}^+ \geq \hat{c}_{n-1}^- x^{B(X_n)}$. The following holds

$$\mathbb{1}(\overbrace{\hat{r}_{n-1}^+ \geq \hat{c}_{n-1}^- x^{B(X_n)}}^{A(n)}) + \mathbb{1}(\overbrace{c^*X_n < \hat{r}_{n-1}^+(X_n) < \hat{c}_{n-1}^- x^{B(X_n)}}^{\mathcal{E}_2}) + \mathbb{1}(\overbrace{\hat{r}_{n-1}^+(X_n) \leq c^*X_n \leq r(X_n)}^{\mathcal{E}_4})$$

$$= \mathbb{1}(\underbrace{r(X_n) \geq c^*X_n}_{\mathcal{E}_0}) + \mathbb{1}(\underbrace{c^*X_n \geq \hat{r}_{n-1}^+(X_n) \geq \hat{c}_{n-1}^- x^{B(X_n)}}_{\mathcal{E}_1}) + \mathbb{1}(\underbrace{\hat{r}_{n-1}^+(X_n) > c^*X_n > r(X_n)}_{\mathcal{E}_3}).$$

To prove it, just notice the followings:

- $A(n) \cap \mathcal{E}_4 = \mathcal{E}_0 \cap \mathcal{E}_1$,
- $\mathcal{E}_2$ is disjoint with both $A(n)$ and $\mathcal{E}_4$,
- $\mathcal{E}_3$ is disjoint with both $\mathcal{E}_0$ and $\mathcal{E}_1$,
- $A(n) \cup \mathcal{E}_2 \cup \mathcal{E}_4 = \mathcal{E}_0 \cup \mathcal{E}_1 \cup \mathcal{E}_3$.

It then comes

$$\mathbb{1}(A(n)) + \mathbb{1}(\mathcal{E}_2) + \mathbb{1}(\mathcal{E}_4) = \mathbb{1}(A(n) \cup \mathcal{E}_2 \cup \mathcal{E}_4) + \mathbb{1}(A(n) \cap \mathcal{E}_4)$$
$$= \mathbb{1}(\mathcal{E}_0 \cup \mathcal{E}_1 \cup \mathcal{E}_3) + \mathbb{1}(\mathcal{E}_0 \cap \mathcal{E}_1)$$
$$= \mathbb{1}(\mathcal{E}_0) + \mathbb{1}(\mathcal{E}_1) + \mathbb{1}(\mathcal{E}_3).$$

The first equality comes from the second point; the second from the first and last point; while the last equality comes from the third point. This gives the following

$$\mathbb{1}(A(n)) = \mathbb{1}(\mathcal{E}_0) + \mathbb{1}(\mathcal{E}_1) - \mathbb{1}(\mathcal{E}_2) + \mathbb{1}(\mathcal{E}_3) - \mathbb{1}(\mathcal{E}_4).$$

The quantity of interest is then rewritten as:

$$(c^* - \hat{\gamma}_n)\,\mathbb{E}\left[X_n\,\mathbb{1}(A(n)) + S_n\right] = c^*\,\mathbb{E}\left[X_n\,\mathbb{1}(A(n)) + S_n\right] - \mathbb{E}\left[r(X_n)\,\mathbb{1}(A(n))\right]$$
$$= c^*\,\mathbb{E}\left[X_n\big(\mathbb{1}(\mathcal{E}_1) - \mathbb{1}(\mathcal{E}_2) + \mathbb{1}(\mathcal{E}_3) - \mathbb{1}(\mathcal{E}_4)\big)\right]$$

$$- \mathbb{E}\left[r(X_n)\big(\mathbb{1}(\mathcal{E}_1) - \mathbb{1}(\mathcal{E}_2) + \mathbb{1}(\mathcal{E}_3) - \mathbb{1}(\mathcal{E}_4)\big)\right]$$
$$\leq c^* \, \mathbb{E}\left[X_n\big(\mathbb{1}(\mathcal{E}_1) + \mathbb{1}(\mathcal{E}_3)\big)\right] - \mathbb{E}\left[r(X_n)\big(\mathbb{1}(\mathcal{E}_1) - \mathbb{1}(\mathcal{E}_2) + \mathbb{1}(\mathcal{E}_3) - \mathbb{1}(\mathcal{E}_4)\big)\right]$$

Let us now bound the last four terms.

1. Recall that $\mathcal{E}_1 = \{c^* X_n \geq \hat{r}_{n-1}^+(X_n) \geq \hat{c}_{n-1}^- x^{B(X_n)}\}$, so that

$$\begin{aligned}
\mathbb{E}[r(X_n)\,\mathbb{1}(\mathcal{E}_1)] &= \mathbb{E}[\hat{r}_{n-1}^+(X_n)\,\mathbb{1}(\mathcal{E}_1)] + \mathbb{E}[(r(X_n) - \hat{r}_{n-1}^+(X_n))\,\mathbb{1}(\mathcal{E}_1)] \\
&\geq \mathbb{E}[\hat{c}_{n-1}^- x^{B(X_n)}\,\mathbb{1}(\mathcal{E}_1)] - \mathbb{E}[|r(X_n) - \hat{r}_{n-1}^+(X_n)|\,\mathbb{1}(\mathcal{E}_1)] \\
&\geq c^* \, \mathbb{E}[x^{B(X_n)}\,\mathbb{1}(\mathcal{E}_1)] - C\,\mathbb{E}[|\hat{c}_{n-1}^- - c^*|\,\mathbb{1}(\mathcal{E}_1)] - \mathbb{E}[|r(X_n) - \hat{r}_{n-1}^+(X_n)|\,\mathbb{1}(\mathcal{E}_1)] \\
&\geq c^* \, \mathbb{E}[X_n\,\mathbb{1}(\mathcal{E}_1)] - C\,\mathbb{E}[|\hat{c}_{n-1}^- - c^*|\,\mathbb{1}(\mathcal{E}_1)] - \mathbb{E}[|r(X_n) - \hat{r}_{n-1}^+(X_n)|\,\mathbb{1}(\mathcal{E}_1)] + c^* h \mathbb{P}(\mathcal{E}_1).
\end{aligned}$$

2. $\mathcal{E}_2$ happens with probability at most $n\delta$ since $x^{B(X_n)} \leq X_n$ and using Proposition 4.4, which upper bounds the second term by $Dn\delta$.

3. Recall that $\mathcal{E}_3 = \{\hat{r}_{n-1}^+(X_n) > c^* X_n > r(X_n)\}$ so that the third term is bounded as

$$\begin{aligned}
\mathbb{E}[r(X_n)\,\mathbb{1}(\mathcal{E}_3)] &= \mathbb{E}[\hat{r}_{n-1}^+(X_n)\,\mathbb{1}(\mathcal{E}_3)] + \mathbb{E}[(r(X_n) - \hat{r}_{n-1}^+(X_n))\,\mathbb{1}(\mathcal{E}_3)] \\
&\geq c^* \, \mathbb{E}[X_n\,\mathbb{1}(\mathcal{E}_3)] - \mathbb{E}[|r(X_n) - \hat{r}_{n-1}^+(X_n)|\,\mathbb{1}(\mathcal{E}_3)].
\end{aligned}$$

4. $\mathcal{E}_4$ happens with probability at most $n\delta$ thanks to Lemma 4.2, which upper bounds the fourth term by $Dn\delta$.

Putting everything together we get

$$\begin{aligned}
(c^* - \hat{\gamma}_n)\,\mathbb{E}\left[X_n\,\mathbb{1}(A(n)) + S_n\right] \leq\ & \mathbb{E}[|r(X_n) - \hat{r}_{n-1}^+(X_n)|\,\mathbb{1}(\mathcal{E}_1 \cup \mathcal{E}_3)] + C\,\mathbb{E}[|\hat{c}_{n-1}^- - c^*|\,\mathbb{1}(\mathcal{E}_1)] \\
& + 2Dn\delta + c^* h.
\end{aligned}$$

The result follows by noting that on $\mathcal{E}_1$, with probability at least $1 - n\delta$, then $c^* - 2\xi_{n-1} \leq \hat{c}_{n-1}^-$, and similarly for $\hat{r}_{n-1}^+ - r$ on $\mathcal{E}_1 \cup \mathcal{E}_3$. $\qquad\square$

# F Proofs of Appendix A

## F.1 Proofs of Appendices A.1 and A.2

*Proof of Theorem A.1.* The proof of this theorem is almost a direct consequence of the proofs of Proposition E.1 and Theorem 4.5, it only requires a few tweaks.

Similarly to Lemma 4.2, it can be shown that $|\hat{r}_n(x) - r(x)| \leq \sigma\sqrt{\frac{\ln(K/\delta)}{2N_{\{x\}}}}$ with probability at least $1 - 2N\delta$ for all $n \leq N$ and $x$. The uncertainty on $\hat{c}_n$ is shown similarly, except for the term $\mathbb{E}[|\hat{r}_n(X_n) - r(X_n)|]$ which is bounded as follows:

$$
\begin{aligned}
\mathbb{E}[|\hat{r}_n(X_n) - r(X_n)|] &= \mathbb{E}\left[\sum_x p(x)\,\mathbb{E}[|\hat{r}_n(X_n) - r(X_n)| \mid N_{\{x\}}]\right] \\
&\leq \mathbb{E}\left[\sum_x p(x)\min\left(\frac{\sigma}{2\sqrt{N_{\{x\}}}}, D - E\right)\right] \quad \text{using Hoeffding's inequality} \\
&\leq \sum_x p(x)\left(\frac{\sigma}{\sqrt{2p(x)n}} + (D-E)e^{-\frac{p(x)n}{8}}\right) \\
&\leq \sigma\sqrt{\frac{K}{2n}} + \frac{8K(D-E)}{n}.
\end{aligned}
$$

The Chernoff's bound $\mathbb{P}(N_{\{x\}} \leq \frac{p(x)}{2n}) \leq e^{-\frac{p(x)n}{8}}$ yields the second inequality; while the third one comes from Cauchy-Schwarz inequality and uses that $pe^{-\frac{pn}{8}} \leq \frac{8}{n}$.

Following the same arguments and as standard in multi-armed bandit, we basically need to compute the number of times tasks $x$ are incorrectly accepted. Consider the event where, for all tasks, it holds $\hat{r}_n^+(x) \leq r(x) + 2(D-E)\sqrt{\frac{\ln(K/\delta)}{N(x)}}$ (with the standard convention that $1/0 = +\infty$) and that $\hat{c}_n^- \geq c^* - 4\lambda(D-E)\sqrt{K\frac{\ln(1/\delta)}{n}}$. Then, for $n$ such that $r(x) + 2\eta_n(x) < (c^* - 2\xi_n)x$, $x$ stops being accepted. In particular, this yields for some constant $\kappa'$ that the total number of accepted tasks of duration $x$ (on this event) is smaller than

$$
N_{\{x\}} \leq \kappa'\frac{K\lambda^2(C^2 + (D-E)^2) + (\sigma^2 + (D-E)^2)(1 + \lambda^2 C^2)\ln(K/\delta)}{(c^*x - r(x))^2} + \sqrt{K},
$$

which gives a contribution to the regret of the order of (up to a multiplicative constant)

$$
\frac{K\lambda^2(C^2 + (D-E)^2) + (\sigma^2 + (D-E)^2)(1 + \lambda^2 C^2)\ln(K/\delta)}{c^*x - r(x)} + \sqrt{K}(c^*x - r(x)),
$$

We conclude as usual thanks to the choice of $\delta$ that ensures that the contribution to the regret on the complimentary event is negligible. $\square$

*Proof of Theorem A.4.* Similarly to the proof of Theorem 4.5, denoting $\Delta(x) = c^*x - r(x)$, the regret can be decomposed as

$$
\begin{aligned}
R(T) &\leq \mathbb{E}\sum_{n=1}^{\theta}\sum_{j=1}^{M}\mathbb{1}(x \in B_j)\Delta(x)\,\mathbb{1}(2\eta_{n-1}(x) + 2\xi_{n-1}C \geq \Delta(x) \geq 0) + 4nD\delta \\
&\leq \mathbb{E}\sum_{n=1}^{\theta}\sum_{j=1}^{M}\mathbb{1}(x \in B_j)\Delta(x)\,\mathbb{1}(4\eta_{n-1}(x)C \geq \Delta(x) \geq 0) + \mathbb{1}(x \in B_j)\Delta(x)\,\mathbb{1}(4\xi_{n-1}C \geq \Delta(x) \geq 0) + 4nD\delta
\end{aligned}
$$

$$\text{(F.1)}$$

The contribution of the third term can be bounded similarly to Theorem 4.5 and is $\mathcal{O}(1)$.

Using the margin condition, the second term scales with $\sum_{n=1}^{\lambda T+1}(C\xi_{n-1})^{1+\alpha}$, which is of order

$$
(\lambda CL^{1-\frac{2}{2\beta+1}})^{1+\alpha}(\lambda T + 1)^{1-\frac{\beta}{2\beta+1}(1+\alpha)}
$$

.

It now remains to bound the first term. It can be done using the analysis of Perchet and Rigollet (2013), which we sketch here.

The idea is to divide the bins into two categories for some constant $c_1$ scaling with $\sigma C^{\frac{2\beta+1}{2}} L \sqrt{\log(\lambda T + 1)}$:

- *well behaved* bins, for which $\exists x \in B, \Delta(x) \geq c_1 M^{-\beta}$,

- *ill behaved* bins, for which $\forall x \in B, \Delta(x) < c_1 M^{-\beta}$.

The first term in Equation (F.1) for ill behaved bins is directly bounded, using the margin condition, by a term scaling with

$$c_1^{1+\alpha} M^{-\beta(1+\alpha)}(\lambda T + 1) \approx \sigma^{1+\alpha} C^{\frac{1+\alpha}{2}} L^{\frac{1+\alpha}{2\beta+1}} \log(\lambda T + 1)^{\frac{1+\alpha}{2}} (\lambda T + 1)^{1 - \frac{\beta}{2\beta+1}(1+\alpha)}.$$

Now denote $\mathcal{J} \subset \{1, \ldots, M\}$ the set of well behaved bins and for any $j \in \mathcal{J}, \Delta_j = \max_{x \in B_j} \Delta(x)$. Using classical bandit arguments, the event $\{x \in B_j\} \cap \{4\eta_{n-1}(x)C \geq \Delta(x) \geq 0\}$ only holds at most $\kappa''(\sigma^2 + L^2 \left(\frac{C}{M}\right)^2)\frac{\log(\lambda T)}{\Delta_j^2}$ times.

Assuming w.l.o.g. that $\mathcal{J} = \{1, \ldots, j_1\}$ and $\Delta_1 \leq \ldots \leq \Delta_{j_1}$, the margin condition can be leveraged to show that $\Delta_j \geq \left(\frac{\kappa j}{\kappa_0 M}\right)^{\frac{1}{\alpha}}$. The contribution of well behaved bins then scales as

$$(\sigma^2 + L^2 \left(\frac{C}{M}\right)^2) \sum_{j=1}^{j_1} \frac{\log(\lambda T + 1)}{\Delta_j} \leq \sigma^2 \log(\lambda T + 1) \left( \frac{j_2 M^\beta}{c_1} + \sum_{j=j_2+1}^{j_1} \left(\frac{j}{M}\right)^{-\frac{1}{\alpha}} \right), \quad \text{(F.2)}$$

where $j_2$ is some integer of order $c_1^\alpha M^{1-\alpha\beta}$ so that for any $j \leq j_2, c_1 M^{-\beta} \geq \left(\frac{\kappa j}{\kappa_0 M}\right)^{\frac{1}{\alpha}}$.

The first term of Equation (F.2) can then be bounded by

$$(\sigma^2 + L^2 \left(\frac{C}{M}\right)^2)c_1^{\alpha-1} M^{1+\beta(1-\alpha)} \log(\lambda T + 1) \lesssim \sigma^{1+\alpha} C^{\frac{1+\alpha}{2}} L^{\frac{1+\alpha}{2\beta+1}} (\lambda T + 1)^{1 - \frac{\beta}{2\beta+1}(1+\alpha)} \log(\lambda T + 1)^{\frac{1+\alpha}{2}},$$

where $\lesssim$ means that the inequality holds up to some universal constant $\kappa'$.

The sum in Equation (F.2) can be bounded as follows:

$$\sum_{j=j_2+1}^{j_1} \left(\frac{j}{M}\right)^{-\frac{1}{\alpha}} \leq \int_{\frac{j_2}{M}}^{1} x^{\frac{1}{\alpha}} dx$$

$$\lesssim \frac{c_1^{\alpha-1} M^{\beta(1-\alpha)}}{1-\alpha}.$$

Finally, the first term of Equation (F.1) scales as

$$\sigma^{1+\alpha} C^{\frac{1+\alpha}{2}} L^{\frac{1+\alpha}{2\beta+1}} (\lambda T + 1)^{1 - \frac{\beta}{2\beta+1}(1+\alpha)} \log(\lambda T + 1)^{\frac{1+\alpha}{2}},$$

which finally yields Theorem A.4 when gathering everything. $\qquad \square$

## F.2 Proofs of Appendix A.3

The following Lemma indicates that with arbitrarily high probability, our upper/lower estimations are correct

**Lemma F.1.** *With a constant $\kappa_6$ independent from $n$, the events*

$$|p_n(s) - p(s)| \, \mathbb{E}[X \, \mathbb{1}(X \geq s) + \frac{1}{\lambda}] \leq \zeta_n + \frac{\kappa_6}{n}.$$

*hold with probability at least $1 - \delta$, simultaneously for all $n \in \{1, \ldots, S\}$ and all $s \in [s_n, C]$.*

*Proof.* Let

$$p(s) = \frac{\lambda \, \mathbb{E}[r(X) \, \mathbb{1}(X \geq s)]}{1 + \lambda \, \mathbb{E}[X \, \mathbb{1}(X \geq s)]} =: \frac{\nu(s)}{\mu(s)}. \tag{F.3}$$

and

$$p_n(s) = \frac{\lambda \frac{1}{n} \sum_{i=1}^{n} Y_i \, \mathbb{1}(X_i \geq s)}{1 + \lambda \frac{1}{n} \sum_{i=1}^{n} X_i \, \mathbb{1}(X_i \geq s)} =: \frac{\nu_n(s)}{\mu_n(s)}. \tag{F.4}$$

Let us rewrite

$$\frac{\mu(s)}{\lambda} = \frac{1}{\lambda} + s(1 - F(s)) + \int_s^C 1 - F(x) dx \text{ and}$$

$$\frac{\mu_n(s)}{\lambda} = \frac{1}{\lambda} + s(1 - F_n(s)) + \int_s^C 1 - F_n(x) dx,$$

where $F_n$ is the empirical distribution function $F_n(x) := \frac{1}{n} \sum_{i=1}^{n} \mathbb{1}(X_i \leq x)$. Hence

$$\frac{|\mu(s) - \mu_n(s)|}{\lambda} \leq s|F(s) - F_n(s)| + \int_s^C |F(x) - F_n(x)| dx$$

$$\leq s|F(s) - F_n(s)| + (C - s) \max_{x \in [s, C]} |F(x) - F_n(x)|$$

$$\leq C \max_{x \in [s, C]} |F(x) - F_n(x)|.$$

The Dvoretzky–Kiefer–Wolfowitz (DKW) inequality ensures that the event $|\mu(s) - \mu_n(s)| \leq \lambda C \sqrt{\frac{1}{2n} \ln \left( \frac{2S}{\delta} \right)}$, $\forall s \in [0, C]$, holds with probability at least $1 - \frac{\delta}{S}$.

Denote by $s^1, s^2, \ldots, s^n$ a permutation of observed task durations $X_1, \ldots, X_n$ such that $s^1 \leq s^2 \leq \cdots \leq s^n$. Also define $s^0 = s_n$ for completeness. Let $s \in [s^k, s^{k+1})$, since $E \leq 0$, it comes

$$\nu(s) = \lambda \, \mathbb{E}[r(X) \, \mathbb{1}(X \geq s)]$$
$$\leq \nu(s^k) - \lambda E(F(s) - F(s^k))$$
$$\leq \nu(s^k) - \lambda E(F(s^{k+1}) - F(s^k))$$
$$= \nu(s^k) - \lambda E\left( F(s^{k+1}) - F_n(s^{k+1}) + F_n(s^{k+1}) - F_n(s^k) + F_n(s^k) - F(s^k) \right)$$

Hence,

$$\nu(s) \leq \nu(s^k) - 2\lambda E \sqrt{\frac{1}{2n} \ln \left( \frac{S}{\delta} \right)} - \frac{\lambda E}{n} \quad \forall k \in \{1, \ldots, n\},$$

holds as soon as the DKW inequality above holds. We prove analogously for the event

$$\nu(s) \geq \nu(s^k) - 2\lambda D \sqrt{\frac{1}{2n} \ln \left( \frac{S}{\delta} \right)} - \frac{\lambda D}{n} \quad \forall k \in \{1, \ldots, n\}.$$

Recall that $s^k = X_{i_k}$ for some $i_k$ for $k \in \{1, \ldots, n\}$. By Hoeffding's inequality with $(X_i)_{i \neq i_k}$ as samples and a union bound on $\{0, \ldots, n\}$, the event

$$|\nu(s^k) - \nu_n(s^k)| \leq \lambda \sqrt{\sigma^2 + \frac{(D - E)^2}{4}} \sqrt{\frac{1}{2(n-1)} \ln \left( \frac{2S(n+1)}{\delta} \right)} \quad \forall k \in \{0, \ldots, n\}$$

holds with probability at least $1 - \frac{\delta}{S}$.

Putting everything together with the fact that $\nu_n(s) = \nu_n(s^k)$ and $n \leq S$, one gets that the event

$$|\nu_n(s) - \nu(s)| \leq \lambda \left( \sqrt{\sigma^2 + \frac{(D-E)^2}{4}} + \sqrt{2}(D-E) \right) \sqrt{\frac{\ln(2S/\delta)}{n-1}} + \frac{\lambda(D-E)}{n}$$

holds with probability at least $1 - \frac{\delta}{S}$ for all $s \in [s_n, C]$ if the DKW inequality also holds.
Furthermore,

$$\left| \frac{\nu_n(s)}{\mu_n(s)} - \frac{\nu(s)}{\mu(s)} \right| \leq |\nu_n(s)| \left| \frac{\mu(s) - \mu_n(s)}{\mu(s)\mu_n(s)} \right| + \frac{1}{\mu(s)} |\nu_n(s) - \nu(s)| \tag{F.5}$$

$$\leq \frac{\left( \lambda(D-E) + |\nu_n(s) - \nu(s)| \right) |\mu(s) - \mu_n(s)| + |\nu_n(s) - \nu(s)|}{1 + \lambda \mathbb{E}[X \mathbb{1}(X \geq s)]}. \tag{F.6}$$

Hence with probability at least $1 - \frac{2\delta}{S}$ one has for all $s \in [s_n, C]$,

$$\begin{aligned}
|p_n(s) - p(s)| \, \mathbb{E}[X \mathbb{1}(X \geq s) + \frac{1}{\lambda}] \leq & \lambda \left( \sqrt{\sigma^2 + \frac{(D-E)^2}{4}} + \frac{D-E}{\sqrt{2}}(\lambda C + 2) \right) \sqrt{\frac{\ln(2S/\delta)}{n-1}} \\
& + \lambda \frac{D-E}{n} \\
& + \lambda C \left( \sqrt{\frac{\sigma^2}{2} + \frac{(D-E)^2}{8}} + D - E \right) \frac{\ln\left( \frac{2S}{\delta} \right)}{n-1} \\
& + \lambda C \frac{D-E}{\sqrt{2}n^{3/2}} \sqrt{\ln\left( \frac{2S}{\delta} \right)}.
\end{aligned}$$

We conclude using a union bound over $\{1, \ldots, S\}$. $\qquad\square$

The goal of the next lemma is to ensure that the sequence $(s_n)_{1 \leq n \leq S}$ is indeed bellow the the optimal threshold $s^*$ with high probability.

**Lemma F.2.** *With probability at least $1 - \delta$, the events*

$$s_n \leq s^* \quad and \quad 0 \leq (p_n(s_n^*) - p_n(s^*)) \left( \frac{1}{\lambda} + \frac{1}{n} \sum_{i=1}^{n} X_i \mathbb{1}(X_i \geq s) \right) \leq 2\zeta_n,$$

*hold simultaneously in all stages $n \in \{1, \ldots, S\}$.*

*Proof.* The proof is by induction on $n \geq 1$. For $n = 1$, $s_1 = 0 \leq s^*$. We prove in a similar manner to Lemma F.1, that the events

$$(p(s) - p_n(s)) \left( \frac{1}{\lambda} + \frac{1}{n} \sum_{i=1}^{n} X_i \mathbb{1}(X_i \geq s) \right) \leq \zeta_n \quad \forall s \in [s_n, C] \tag{F.7}$$

simultaneously hold for all stages $n \in \{1, \ldots, S\}$, with probability at least $1 - \delta$. The only difference with Lemma F.1 is that the term $|\nu_n(s) - \nu(s)|$ in Equation (F.6) does not appear here, removing the $\frac{\kappa_6}{n}$ term in Lemma F.1. In particular, this implies the events

$$(p(s^*) - p_1(s^*)) \left( \frac{1}{\lambda} + X_i \mathbb{1}(X_1 \geq s) \right) \leq \zeta_1$$

$$\text{and } (p_1(s_1^*) - p(s_1^*)) \left( \frac{1}{\lambda} + X_1 \mathbb{1}(X_1 \geq s) \right) \leq \zeta_1$$

Since $p(s_1^*) \leq p(s^*)$, one has that

$$(p_1(s_1^*) - p_1(s^*)) \left( \frac{1}{\lambda} + X_i \mathbb{1}(X_1 \geq s) \right) \leq 2\zeta_1$$

holds when Equation (F.7) holds.

Now for any $n > 1$, the induction assumption implies that $s^* \in \mathcal{S}_n$, hence $s^* \geq s_n$. The rest of the induction follows the steps of the base case $n = 1$. Finally, one has $0 \leq p_n(s_n^*) - p_n(s^*)$ by definition of $s_n^*$. $\qquad\square$

The following proposition allows to control the regret in stage $n$.

**Proposition F.3.** *For some constant $\kappa_7$ independent from $n$, with probability at least $1 - \delta$, the events*

$$(p(s^*) - p(s_n)) \, \mathbb{E}[X \, \mathbb{1}(X \geq s_n) + \frac{1}{\lambda}] \leq 4\zeta_{n-1} + \frac{\kappa_7}{n}.$$

*hold simultaneously in all stages $n \in \{2, \dots, S\}$.*

*Proof.* With probability at least $1 - \delta$, the following inequality hold uniformly for all stages $n \geq 2$ by Lemma F.1:

$$(p(s^*) - p(s_n)) \, \mathbb{E}[X \, \mathbb{1}(X \geq s_n) + \frac{1}{\lambda}] \leq (p(s^*) - p_{n-1}(s_n)) \, \mathbb{E}[X \, \mathbb{1}(X \geq s_n) + \frac{1}{\lambda}] + \zeta_{n-1} + \frac{\kappa_6}{n}$$

Now remark, using the notation of Lemma F.1 that, thanks to the DKW inequality and the fact that $s_n \in \mathcal{S}_n$, under the same probability event,

$$(p_{n-1}(s_{n-1}^*) - p_{n-1}(s_n)) \frac{\mu(s_n)}{\lambda} = (p_{n-1}(s_{n-1}^*) - p_{n-1}(s_n)) \frac{\mu_n(s_n)}{\lambda} + (p_{n-1}(s_{n-1}^*) - p_{n-1}(s_n)) \frac{\mu(s_n) - \mu_n(s_n)}{\lambda}$$

$$\leq 2\zeta_{n-1} + 2\zeta_{n-1} C \sqrt{\frac{\ln(\frac{4S}{\delta})}{2(n-1)}}.$$

The definition of $\mathcal{S}_n$ directly bounds the first term. For the second term, note that the definition of $\mathcal{S}_n$ also bounds $p_{n-1}(s_{n-1}^*) - p_{n-1}(s_n)$. Combined with the concentration on $\mu_n - \mu$, this bounds the second term. It then follows

$$(p(s^*) - p(s_n)) \frac{\mu(s_n)}{\lambda} \leq (p(s^*) - p_{n-1}(s_{n-1}^*)) \, \mathbb{E}[X \, \mathbb{1}(X \geq s_n) + \frac{1}{\lambda}] + 3\zeta_{n-1} + \frac{\kappa_8}{n}$$

$$\leq (p(s^*) - p_{n-1}(s^*)) \, \mathbb{E}[X \, \mathbb{1}(X \geq s_{n-1}) + \frac{1}{\lambda}] + 3\zeta_{n-1} + \frac{\kappa_8}{n}$$

$$\leq 4\zeta_{n-1} + \frac{\kappa_8 + \kappa_6}{n} \text{ by Lemma F.1 and since } s^* \geq s_{n-1} \text{ by Lemma F.2.}$$

$\qquad\square$

**Lemma F.4.** *For $S = 2\lambda T + 1$, we have*

$$\mathbb{E}\left[(\theta - S)_+\right] \leq 4.$$

*Proof.* Similarly to the proof of Theorem 3.3, note that $\theta - 1$ is dominated by a random variable following a Poisson distribution of parameter $\lambda T$. It now just remains to show that for any random variable $Z$ following a Poisson distribution of parameter $\lambda$, it holds that $\mathbb{E}\left[(Z - 2\lambda)_+\right] \leq 4$.

Thanks to Canonne (2017), the cdf of $Z$ can be bounded as follows for any positive $x$:

$$\mathbb{P}(Z \geq \lambda + x) \leq e^{-\frac{x^2}{2(\lambda + x)}},$$

which implies for $x = \lambda + t$ with $t > 0$:

$$\mathbb{P}(Z - 2\lambda \geq t) \leq e^{-\frac{(t+\lambda)^2}{2(2\lambda + t)}}$$

$$\leq e^{-\frac{t+\lambda}{4}}.$$

From there, the expectation of $(Z - 2\lambda)_+$ can be directly bounded:

$$\mathbb{E}[(Z - 2\lambda)_+] \leq \int_0^\infty e^{-\frac{t+\lambda}{4}} \, \mathrm{d}t$$

$$\leq 4.$$

This concludes the proof. $\qquad\square$

*Proof of Theorem A.6.* Here we have

$$\theta = \min\{n \in \mathbb{N} \mid \sum_{i=1}^{n} S_i + X_i \mathbb{1}(X_i \geq s_i) > T\}.$$

Algorithm 3 yields the regret

$$R(T) = c^* T - \mathbb{E}\left[\sum_{i=1}^{\theta} r(X_i) \mathbb{1}(X_i \geq s_i)\right]$$

$$= c^* T - \mathbb{E}\left[\sum_{i=1}^{\theta} \mathbb{E}\left[r(X_i) \mathbb{1}(X_i \geq s_i)\right]\right] \quad \text{by Wald's equation}$$

$$= c^* T - \mathbb{E}\left[\sum_{i=1}^{\theta} p(s_i) \mathbb{E}\left[X_i \mathbb{1}(X_i \geq s_i) + S_i\right]\right]$$

$$\leq \mathbb{E}\left[\sum_{i=1}^{\theta} (c^* - p(s_i)) \mathbb{E}\left[X_i \mathbb{1}(X_i \geq s_i) + S_i\right]\right].$$

Note that conversely to the previous sections, when using Wald's equation the expectation here is taken conditionally to threshold $s_i$. Bounding separately the first two terms and the whole sum for small probability events, Proposition F.3 yields

$$R(T) \leq \sum_{n=3}^{S} \left(4\zeta_{n-1} + \frac{\kappa_7}{n}\right) + 2(D - E) + (D - E)(S\delta + \mathbb{E}\left[(\theta - S)_+\right])$$

$$\leq 8 \left(\sqrt{\sigma^2 + \frac{(D - E)^2}{4}} + \frac{D - E}{\sqrt{2}}(\lambda C + 2)\right) \sqrt{S \ln\left(\frac{2(S + 1)}{\delta}\right)}$$

$$+ (4\lambda(D - E) + \kappa_7)\ln(eS) + (D - E)(2 + S\delta + \mathbb{E}\left[(\theta - S)_+\right])$$

Using the given values for $S, \delta$ and Lemma F.4 finally yields Theorem A.6. $\qquad\square$

## F.3   Proofs of Appendix A.4

*Proof of Proposition A.8.*
1) We first prove for all algorithms in Section 4 and Appendix A, when the reward function is unknown and noisy observations are observed.

Following the same lines of the proof of Theorem 4.5, we first write the regret in a bandit form:

$$R(T) \geq \mathbb{E}\left[\sum_{n=1}^{\theta} \mathbb{E}\left[\left(r(X_n) - c^* X_n\right)\left(\mathbb{1}(r(X_n) \geq c^* X_n) - \mathbb{1}(A(n))\right)\right]\right] - c^* C. \qquad \text{(F.8)}$$

Now consider the one arm contextual bandit where, given a context $X$, the normalized reward is $c^* X$ and the arm returns a noised reward of mean $r(X)$. The sum in Equation (F.8) exactly is the regret incurred by the strategy $A$ in this one armed contextual bandit problem, with horizon $\theta$.

Although $\theta$ is random and depends on the strategy of the agent, it is larger than $\frac{\lambda T}{1+C}$ with probability at least $\alpha > 0$, constant in $T$. Moreover, the one armed contextual bandit problem is easier than the agent problem, as $c^*$ is known only in the former. Thanks to these two points, $R(T)$ is larger than

$$R(T) \geq \alpha \tilde{R}\left(\frac{\lambda T}{1 + C}\right) - c^* C$$

where $\tilde{R}$ is the regret in this one armed contextual bandit problem. Proposition A.8 then follows from classical results in contextual bandits (see, e.g., Audibert and Tsybakov, 2007; Rigollet and Zeevi, 2010).

Note here that we only consider a subclass of all the one armed contextual bandit problems. Indeed, we fixed the normalized reward to $c^* X$ and the arm reward must satisfy $c^* = \lambda \mathbb{E}[(r(X) - c^* X)_+]$. Yet this subclass is large enough and the existing proofs (Audibert and Tsybakov, 2007; Rigollet and Zeevi, 2010) can be easily adapted to this setup.

2) It now remains to prove a $\Omega(\sqrt{T})$ bound when the reward function $r$ is known. Consider the following setting

$$X = \begin{cases} 1 \text{ with proba } \frac{1}{2} + \varepsilon \\ 2 \text{ with proba } \frac{1}{2} - \varepsilon \end{cases}$$

$$\text{and} \quad \begin{cases} r(1) = \frac{1}{2} \\ r(2) = 2. \end{cases}$$

Now consider two worlds where $\varepsilon = \Delta$ in the first one, and $\varepsilon = -\Delta$ in the second one for some $\Delta > 0$. Basic calculations then give the following

$$c_1^* = \frac{1}{2} - \frac{2\Delta}{5} + o(\Delta) \quad \text{in the first world,}$$

$$c_2^* = \frac{1}{2} + \frac{\Delta}{2} + o(\Delta) \quad \text{in the second world.}$$

The optimal strategy then accepts all tasks in the first world, while it only accepts tasks of duration 2 in the second one. Classical lower bound techniques (see, e.g., Lattimore and Szepesvári, 2020, Theorem 14.2) then show that for $\Delta = \frac{1}{\sqrt{\lambda T}}$, with some constant probability and positive constant $\alpha$ both independent from $T$, any strategy

- either rejects $\alpha \lambda T$ tasks $X_t = 1$ in the first world,

- or accepts $\alpha \lambda T$ tasks $X_t = 1$ in the second world after receiving $\lambda T$ task propositions.

In any case, this means that any strategy has a cumulative regret of order $\sqrt{T}$ in at least one world. $\qquad \square$