# OpenReview forum: "Making the most of your day: online learning for optimal allocation of time"
_NeurIPS.cc/2021/Conference — NeurIPS 2021 Poster_

### Official Review · Reviewer_yYEd · 2021-07-12

**Rating:** 6
**Confidence:** 3

**Summary:**

This paper studies an online scheduling problem in which tasks arrive online each with a time and reward. Upon accepting a task, the agent becomes busy for a certain duration and can only start taking new tasks when finishing the previous task. The authors consider the case where the agent knows the reward function but not the duration distribution, and where the agent also does not know both the reward function. In both cases, the authors propose online learning policies with sublinear regret. The key of the algorithm is the learning of a bang-per-buck threshold above which the algorithm will accept the task. When the rewards are noisy and unknown, the algorithm applies a variant of a regressogram and uses the upper-estimate of the reward function and a lower-estimate of the threshold to decide whether to accept a task. Finally, simulations are conducted on several toy examples to evaluate the performances of different algorithms.

**Limitations And Societal Impact:**

Yes.

**Main Review:**

Overall the concept of time as a resource in online learning is interesting both from a theoretical and applicable-to-the-real-world point of view. As the authors described in the introduction, there are numerous practical scenarios that fit into this model. Technique-wise, the algorithm idea of learning the best threshold is not particularly innovative but is still a natural and reasonable idea. When the reward function is unknown to the agent, the algorithm cleverly modifies the basic algorithm to incorporate non-parametric estimation of $r$ and eventually leads to a nontrivial regret guarantee.

The experiment section is somewhat underwhelming. The algorithms are only tested on toy examples with very simple reward functions. It would certainly strengthen the paper if the authors could find some meaningful case studies and test the algorithms on real-world datasets.

Conceptually, the problem being studied bears a resemblance to the knapsack problem, where the duration of a task corresponds to the weight of an item in the knapsack problem. There are also different variants of online knapsack problems in the literature. This paper also lacks citation and discussion in the related topic.



POST AUTHOR RESPONSE:
Thank you for your response. My overall opinion of the paper remains the same.

**Time Spent Reviewing:**

6 hours

---

> ### Author Response · Authors · 2021-08-09
> **Answer to Reviewer yYEd**
>
> - This work is mainly theoretical and we thus considered simple simulations to test our algorithms, as is usually done in the bandits literature. Real world data analysis is left for future developments and requires much more work such as finding clean and usable data, adapting the model to the specificities of the application (additional assumptions might be considered) and comparing with some already good baseline.
> - We only mentioned bandits with knapsacks as this is the knapsack problem that is closest to our work. Additional references and a discussion on online knapsack will be added in the new version. In particular, the competitive analysis approach of the online knapsack problem might be interesting for the case of adversarial rewards/durations (see our answer to reviewer bRNB).
> Please do not hesitate to point us to references that may be particularly interesting/related.

---

### Official Review · Reviewer_SRSX · 2021-07-12

**Rating:** 4
**Confidence:** 4

**Summary:**

This paper studies a decision making problem of accepting or rejecting of tasks that arrive randomly (wrt to Poisson process) and take random (unknown) durations to complete. By accepting/denying a task, the controller tries to minimizing the expected regret over time. Under this setup, the paper proposed threshold-based decision algorithms first with known reward function then with bandit feedback.

**Limitations And Societal Impact:**

The authors briefly mention the limitations on considering only one resource limitation. In addition, this setting does not consider scenarios where task durations are unknown before making decisions.

**Main Review:**

Originality:
(1) There have been closely related works on reward rate maximization for unknown service-time and reward distributions (Cayci et al. Sigmetrics/POMACS 2019 and Cayci et al. AISTATS 2020).
(2) The authors briefly mention the limitations on considering only one resource limitation. However, there have been works on multiple resource limitations such as (Agrawal et al. 2016).

Quality:  The paper is well written in most parts though, perhaps due to page limitation, it ends abruptly without any conclusion.

Clarity: (1) The equation (2.5) for regret is unclear. Why are we comparing to c* T when the optimal reward may be larger than this (as in Theorem 2.2 we are considering tasks with profitability greater than or equal to c*)?
(2) As the author mentioned, the results of algorithm 1 (Figure 3 and Figure 6) fluctuate a lot and 50 averages are not enough to see a clear trend on its performance.  In addition, I am not sure how boundary effect can be so large that it can give a negative regret.

Significance: Limiting the reward as a function of the task duration (which can be observed before making decision) is a non-trivial assumption. In this way, the setting is extremely similar to contextual bandit. The algorithm takes advantage of this assumption and calculates reward for different task duration bins.

**Time Spent Reviewing:**

3 hours

---

> ### Author Response · Authors · 2021-08-09
> **Answer to Reviewer SRSX**
>
> Originality:
> - Thanks for pointing us to these references: they will be added in the related work section. They are indeed related as they consider profitability maximization. Yet, we want to stress that the considered problem (and solutions) remain very different. Here, the agent first observes a context (the duration) before making her decision. As a consequence, the baseline optimal policy is very different.
> A single arm (accept) is considered here (note that we could easily extend to multiple arms), but the reward of the safe arm (reject) is unknown and depends on the whole distribution of $X$ and the reward function $r$.
> As a consequence, while the goal in the works of Cayci et al. is to determine the arm with the largest reward/time ratio, the main (new) challenge in our work is to determine $c^*$.
> - There might have been a confusion as the term resource was incorrectly used here (this will be clarified in the new version). The mentioned limitation is that the agent can treat only one task at a time. For some applications (e.g., call centers), several tasks can be handled simultaneously. Our analysis cannot be extended to this case, as even computing/describing the optimal policy seems much more intricate. This problem is thus left open for future work and will be further explained in the conclusion.
> On the other hand, our algorithm can easily be extended to the case of multiple limited resources (considered by Agrawal et al.), using classical knapsack bandits ideas.
>
> Quality:
> - There was indeed no conclusion due to the page limitation. As explained in our general answer, a conclusion will be added in the new version.
>
> Clarity:
> - The regret indeed does not compare to the optimal policy, but instead to the optimal static policy, which only differs by a small constant thanks to Equation (2.3). As a consequence, both definitions of regret yield the same bounds and this definition is preferred for simplicity. Note that Cayci et al. also compare with the static optimal policy in their works, for similar reasons.
> We indeed accept tasks with profitability $\geq c^*$, but we also lose some time by remaining idle, waiting for the next task proposal. This is why the optimal cumulative reward is $c^* T$ in the end.
> - The fluctuations of Algo 1 are amplified by the log scale, but are not so significant (of order ~ 100 in Fig. 3). We still observe some fluctuations (due to randomness) and the simulations will be averaged over more runs in the new version, removing these observed fluctuations.
> Also, the regret appears non-negative at times. This is due not only to boundary effects, but also to random fluctuations and the fact that $c^*$ is not computed exactly (the expectation is approximated using Monte Carlo methods). Even a small error in $c^*$ (e.g., $10^{-4}$) might lead to an error in the regret of order 100 (for $T=10^6$). This is not significant for the other algorithms, as their regret is of much larger order. In the new simulations, the estimation of $c^*$ will be refined so that this effect also becomes negligible for Algo 1.
>
> Significance:
> - This problem indeed bears similarities with contextual bandits. It is actually equivalent if $c^*$ is known beforehand. But we want to stress again that the main novelty and difficulty in this problem is the estimation of $c^*$, which depends on the whole distribution of $X$ and the reward function $r$. We propose not only an analytical solution, but also a computationally efficient algorithm as explained in Remarks 3.5, 4.6 and Appendix B.
> The reward only depends on the duration $X$ here, but our work could easily be extended to the case of several possible decisions (arms) or/and multidimensional contexts (i.e., $r$ could depend on $X$ and an additional covariate $Z$), using classical bandits techniques.
>
> Limitations:
> - The duration is here used as a context. If it is not observed beforehand, the problem almost becomes a non-contextual one armed bandits algorithm and the optimal policy is then trivial: it accepts all rides.
> The duration could also be noisy (or equivalently we could observe only $\mathbb{E}[X_i]$), but this does not bring additional difficulties: our algorithms and analyses are also valid for subgaussian noises. We thus ignored this consideration for the sake of clarity.

---

> > ### Author Response · Authors · 2021-08-30
> > **Thanks for your review**
> >
> > Thank you for your comments, and we hope that our answer has addressed your concerns. If you still have concerns after reading our response, please do not hesitate to let us know. We will be happy to answer them.

---

### Official Review · Reviewer_bRNB · 2021-07-16

**Rating:** 7
**Confidence:** 4

**Summary:**

The paper studies a setting where requests arrive according to a Poisson process and an agents must decide to process them or not. If she decides to process the request, she gets busy for an amount of time shown in the request (and cannot process or see the processing times of any requests arriving in this time period) and gets a reward which is a function of the processing time; if not, she remains idle and ready to process new requests. It is assumed that the arrival rate is known but the probability distribution of processing times is not.

The contributions of the paper are as follows:
1) A myopic algorithm with bounded reward difference with respect to the optimal one is derived: it is essentially a threshold policy where the agent accepts a request if the ratio of reward over processing time (profitability) is greater than an appropriately computed threshold.
2)Learning algorithms with regret (with respect to the myopic algorithm) sublinear with the time horizon are derived, in the cases where the reward function is known or not.

The algorithms seem to be based in estimating fast the optimal profitability threshold $c^*$ and, in the bandit case, on estimating the reward function via splitting the interval of possible processing times into bins (and estimating a reward for each bin).

**Limitations And Societal Impact:**

No relevant negative potential social impact.

**Main Review:**

The analysis is nice and rigorous, also the numerical results help illustrate the behavior of the learning algorithm in practice.

The problem examined seems an important one, as it is a simple abstraction of significant problems in e.g. admission control for computing tasks in servers.

Some questions regarding the setting:
- Is it essential that the inter-arrival times of the demands follow a memoryless (exponential) distribution, or the algorithms and guarantees can be extended to more general inter-arrival times?
- Can the main ideas be extended when processing times are not i.i.d. ?

Other comments:
- Why the curve for the average reward when the reward function is known is non-monotonic in T (figure 3) ?
- A conclusion is missing. For instance, it would be nice to have some ideas for further extensions and exploitation of the results.
- Define what $c_n$ is before using it (in Proposition 3.1)
- Point to equation (4.5) for the definition of $\ksi_n$ in algorithm  2.

**Time Spent Reviewing:**

4

---

> ### Author Response · Authors · 2021-08-09
> **Answer to Reviewer bRNB**
>
> - Exponential inter-arrival times are assumed for simplicity. The proof of Proposition 2.1 largely relies on the memoryless assumption and extending it to more general cases is very intricate. Note that this assumption is equivalent to assuming that task proposals follow a Poisson process, which is classically assumed in the models of the mentioned applications.
>
> - Our algorithms cannot be directly extended to non i.i.d. processing times $X_i$, and this problem is left open for future work. The case of adversarial $X_i$ is linked to other online problems, such as knapsack online problem (see our answer to reviewer yYed). In this case, it might be interesting to use a competitive analysis approach.
>
> Other comments:
> - non-monotonicity of regret: see our answer to reviewer SRSX
> - a conclusion will be added in the new version as explained in our general reply.

---

### Author Response · Authors · 2021-08-09
**Global answer**

We first want to thank the reviewers for their insightful comments, which will surely improve the quality of this work.
We also want to stress that our paper is mostly theoretical and aims at providing a simple baseline model that could be used in various possible applications; for each specific application, additional assumptions might be required.
NeurIPS allows an additional page for the camera ready version, which would allow us to add a conclusion, discussing the potential extensions/directions that are left open for future work.

We reply individually to the reviewers’ comments below.

---

### Decision · Program_Chairs · 2021-09-27

**Decision:**

Accept (Poster)

**Comment:**

The reviews are slightly mixed on this work. I also read the paper and found the setup nice and the the writeup is generally clear and well presented.

The authors should take the reviewer suggestions into careful considering, including discussing the related literature. I am satisfied, however, that the novelty in the present work is sufficient to merit publication.